# Flexible motor sequence generation during stereotyped escape responses

**Yuan Wang[1,2†], Xiaoqian Zhang[1,2†], Qi Xin[1,2†], Wesley Hung[3,4], Jeremy Florman[5], Jing Huo[1,2], Tianqi Xu[1,2], Yu Xie[1], Mark J Alkema[5], Mei Zhen[3,4], Quan Wen[1,2,6*]**

[1]Hefei National Laboratory for Physical Sciences at the Microscale, Center for Integrative Imaging, School of Life Sciences, University of Science and Technology of China, Hefei, China; [2]Chinese Academy of Sciences Key Laboratory of Brain Function and Disease, Hefei, China; [3]Samuel Lunenfeld Research Institute, Mount Sinai Hospital, Toronto, Canada; [4]University of Toronto, Toronto, Canada; [5]Department of Neurobiology, University of Massachusetts Medical School, Worcester, United States; [6]Center for Excellence in Brain Science and Intelligence Technology, Chinese Academy of Sciences, Shanghai, China

**Abstract** Complex animal behaviors arise from a flexible combination of stereotyped motor primitives. Here we use the escape responses of the nematode *Caenorhabditis elegans* to study how a nervous system dynamically explores the action space. The initiation of the escape responses is predictable: the animal moves away from a potential threat, a mechanical or thermal stimulus. But the motor sequence and the timing that follow are variable. We report that a feedforward excitation between neurons encoding distinct motor states underlies robust motor sequence generation, while mutual inhibition between these neurons controls the flexibility of timing in a motor sequence. Electrical synapses contribute to feedforward coupling whereas glutamatergic synapses contribute to inhibition. We conclude that *C. elegans* generates robust and flexible motor sequences by combining an excitatory coupling and a winner-take-all operation via mutual inhibition between motor modules.

*For correspondence:
qwen@ustc.edu.cn

†These authors contributed equally to this work

**Competing interests:** The authors declare that no competing interests exist.

## Introduction

Nervous systems transform sensation into a sequence of actions. The motor repertoire, constrained by the biomechanics of gait, comprises a finite number of motor primitives that are stereotyped across individuals (*Ahamed et al., 2019*; *Berman et al., 2014*; *Liu et al., 2018*; *Stephens et al., 2008*). On the other hand, behavioral flexibility allows an animal to explore the action space, and to select better strategies for acquiring rewards or avoiding danger in a changing environment (*Sutton and Barto, 2017*).

Many factors contribute to behavioral flexibility (*Dhawale et al., 2017*; *Gordus et al., 2015*; *Remington et al., 2018a*). Actions may be generated by an inherently noisy system: synapses are unreliable (*Allen and Stevens, 1994*), neurons generate variable spike trains (*Mainen and Sejnowski, 1995*), and neural circuits may operate near the edge of chaos (*van Vreeswijk and Sompolinsky, 1996*). On the other hand, neural networks, whether adaptive or hard-wired, have structures that shape population neural dynamics onto a low dimensional manifold, where nonrandom and ordered activity patterns emerge (*Ganguli et al., 2008*; *Harvey et al., 2012*; *Inagaki et al., 2018*). Computational models have promised to provide a unified view of these observations (*Burak and Fiete, 2012*; *Brennan and Proekt, 2019*; *Mastrogiuseppe and Ostojic, 2018*; *Roberts et al., 2016*), but a deep connection between theories and experiments remains to be established.

The initiation of escape responses of the nematode *Caenorhabdtis elegans* (*C. elegans*) has long been viewed as an instinctive reflex. Upon a gentle touch to its anterior body, the ventral cord-

projecting premotor interneurons AVA/AVD/AVE relay mechanosensory inputs to motor neurons and reliably drive a backward movement (*Chalfie et al., 1985*; *Pirri et al., 2009*; *Wicks et al., 1996*). While *C. elegans* stays committed to its escape decision, the animal remains flexible in its approach to complete the motor sequence. After the reversal, the animal may or may not reorient its body via a deep omega (Ω) turn, before moving forward (*Figure 1A*). This allows the animal to resume forward movement in either the original or a new direction. Notably, which action to select and when to execute exhibit trial to trial variability, and they can be coupled. For example, a

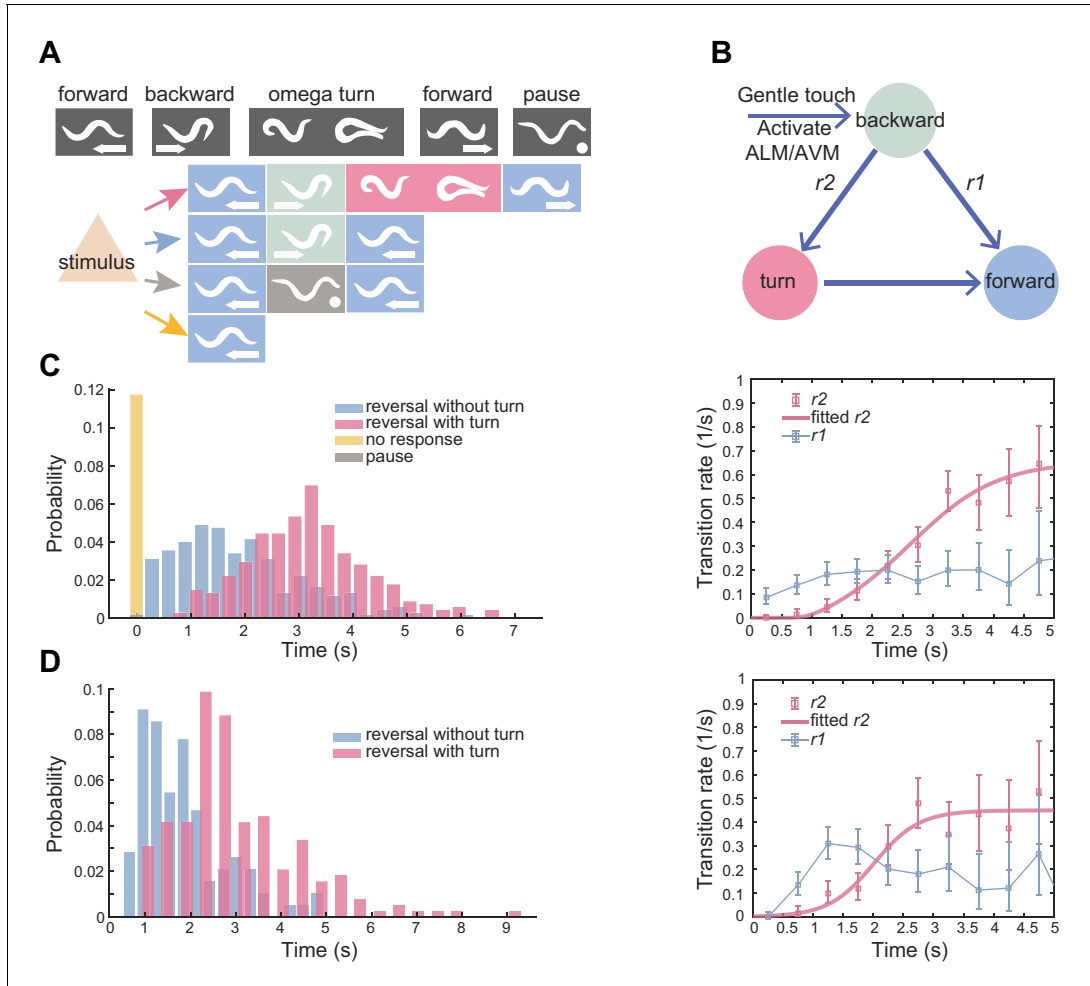

**Figure 1.** Flexible motor sequence generation during *C. elegans* escape responses. (**A**) Optogenetic stimulation of mechanosensory neurons ALM/AVM (P*mec-4*::ChR2) was carried out with blue light (473 nm, 4.63 mW/mm²) for 1.5 s during animal forward movement. Identical stimuli could induce four different behavioral outputs: reversal followed by an omega turn, reversal followed by forward movement, pause state or forward movement (no response). (**B**) Definitions of type-I and type-II transitions. (**C**) Reversal length distribution (left) and transition rates (right) during ALM/AVM (P*mec-4*::ChR2) triggered escape responses. Total number of trials, $n = 674$; reversal without turn, $n = 272$; reversal with turn, $n = 322$; no response, $n = 79$; pause, $n = 1$. Number of animals used can be found in *Supplementary file 4*. Here and below, the error bars of transition rates represent 95% binomial proportion confidence interval. (**D**) Reversal length distribution (left) and transition rates (right) during thermal stimulus induced escape responses. Wild-type, $n = 385$. Animal heads were thermally stimulated by an infrared laser (1480 nm; 400 mA) for 0.75 s. Solid lines are theoretical fits using *Equation 4* and fitted type-II transition rates pass goodness-of-fit test (p>0.05).

The online version of this article includes the following video, source data, and figure supplement(s) for figure 1:

**Source data 1.** Source data for *Figure 1* and *Figure 1—figure supplement 1*.
**Figure supplement 1.** Definition of the type-II transition and statistical test of reversal length distribution.
**Figure 1—video 1.** Three main types of escape responses, Related to *Figure 1*.
https://elifesciences.org/articles/56942#fig1video1

previous study (*Gray et al., 2005*) has shown that a longer reversal is likely to be followed by an omega turn.

We sought to understand algorithms and circuit mechanisms for motor sequence generation by investigating recurrently connected interneurons, which are positioned between sensory neurons and motor neurons in the *C. elegans* nervous system (*Figure 2—figure supplement 1A*). Previous studies on this layer of neural network (*Figure 2—figure supplement 1A*) have implicated their roles in exploratory behaviors (*Gray et al., 2005*; *Iino and Yoshida, 2009*; *Mori and Ohshima, 1995*; *Pierce-Shimomura et al., 1999*). During navigation, *C. elegans* moves towards a new direction by making a reversal and/or a turn in a probabilistic manner. Cell ablation studies revealed that the frequencies of reversals or turns were differentially modulated by many local interneurons including AIB and RIB (*Gray et al., 2005*). Here we ask whether and how activities of local interneurons and their synaptic interactions shape the dynamics of a motor sequence during escape responses.

Several models have been proposed to account for motor sequence generation. In a class of synaptic chain models (*Abeles, 1991*; *Long et al., 2010*; *Xiao et al., 2017*), feedforward excitation between transiently activated groups of neurons controls the timing of actions hierarchically. Sequential neural activity may also emerge from a cooperation between external inputs and local synaptic interactions in a recurrent network (*Rajan et al., 2016*; *Seeds et al., 2014*). We find that neurons encoding distinct motor states, such as reversals and omega turns, use electrical coupling to reliably drive motor state transitions, whereas they exploit mutual inhibition to flexibly control the timing of an action in a sequence. We propose that a form of short-term plasticity in inhibitory synapses contributes to the time-dependent change of transition probability between motor states. Our findings provide new insights into how the nervous system organizes time-ordered and variable motor activities, by which stereotyped and flexible animal behaviors emerge.

## Results

### Stereotypical and flexible motor patterns constitute *C. elegans* escape responses

A potentially threatening sensory stimulus will trigger an animal's escape response. For example, a gentle touch on the *C. elegans* head, which activates specific mechanosensory neurons ALM/AVM (*Chalfie et al., 1985*), can induce a reversal or an omega turn (*Figure 1A* and *Figure 1—video 1*).

We quantitatively characterized the escape responses from transgenic animals in which channelrhodopsin-2 (ChR2) was expressed in ALM/AVM neurons (P*mec-4*::ChR2; *lite*-1), and optogenetic stimulation was given to the same sensory neurons at a defined light intensity and pulse duration during forward movement (see Materials and methods) (*Leifer et al., 2011*). ALM/AVM-triggered backward movements responses were robust (only ~10% trials did not respond, *Figure 1C* left), but subsequent motor sequences constituting each trial varied. Animals exhibited two main types of motor patterns: (1) backward movement was followed by a deep omega turn, and the animal moved forward in a new direction that was different from that before stimulation; (2) an animal executed backward movement and then resumed forward movement in a similar direction as that before stimulation (*Figure 1A* and *Figure 1—video 1*). The head and the tail were diametrically opposed to each other in an omega turn; whereas they were likely aligned to each other in a backward-to-forward movement (*Figure 1—figure supplement 1A*). Occasionally, an animal paused (1/674 trials) before resuming forward movement (*Figure 1A*), which can be regarded as the third motor pattern.

The reversal length distribution is broad (*Figure 1C* left) and likely bimodal (*Figure 1—figure supplement 1B*). This observation motivated us to describe behavior statistics by introducing two types of transitions and the corresponding transition rates $r(t)$. Among all reversals survived to time $t$, $r(t)\Delta t$ computes the fraction of events that will make a transition to another motor state within the time bin $\Delta t$. The type-I (RF) transition rate, $r_1$, determines the transition probability from reversal to forward movement; the type-II (RT) transition rate, $r_2$, determines the transition probability from reversal to omega turn (*Figure 1B* and Materials and methods). $r_1(t)$ rapidly plateaued in about one second, while $r_2(t)$ increased and gradually became the dominant mode (*Figure 1C* right). The escape responses induced by a focused infrared laser light (*Mohammadi et al., 2013*) exhibited qualitatively similar statistics to ALM/AVM-triggered responses (*Figure 1D*).

This quantification, which was consistent with a previous observation and description for spontaneous reversals during exploratory behaviors (*Gray et al., 2005*), confirms the notion that the longer a reversal, the more likely the reversal is followed by a turn.

## Local interneurons in the backward module modulate motor state transitions

We ask how neural dynamics underlie the behavioral variation. Whole brain and multi-neuron calcium imaging of fixed and behaving animals suggested that population interneuron activities, which perform sensorimotor transformation, encode distinct motor states (*Gordus et al., 2015*; *Kato et al., 2015*; *Kawano et al., 2011*; *Li et al., 2014*; *Luo et al., 2014*; *Nguyen et al., 2016*; *Roberts et al., 2016*; *Venkatachalam et al., 2016*; *Figure 2A* and *Figure 2—figure supplement 1B*). Several interneurons, including the ventral-cord-projecting premotor interneurons AVA and AVE, and the local interneurons AIB and RIM, exhibited increased calcium activity during a backward movement (*Kato et al., 2015*; *Laurent et al., 2015*; *Luo et al., 2014*; *Figure 2A* and *Figure 2—figure supplement 1B*).

Structural and functional studies of AIB (*Gray et al., 2005*; *White et al., 1986*) indicate that they may play important roles in motor state transitions (*Figure 2A*). First, AIB establish recurrent connections with the premotor interneurons AVA and AVE that potentiate backward movement either directly through chemical synapses or indirectly through electrical and chemical connections with RIM (*Figure 2A*). Second, AIB form gap junctions with the inter/motor neurons RIV (*White et al., 1986*), which play a role in generating a ventral-biased turning behavior (*Figure 2A*; *Gray et al., 2005*). Third, AIB exhibit ramping calcium activity during reversals (*Kato et al., 2015*; *Laurent et al., 2015*; *Luo et al., 2014*), and finally, laser ablation of AIB significantly reduces the frequency of reversals during food search behavior (*Gray et al., 2005*).

We first examined neuronal correlate of behavioral flexibility in action selection. We compared the AIB ramping activity (P*inx-1*::GCaMP6; P*inx-1*::wCherry) in different action sequences during either spontaneous or thermal-stimulus-triggered behaviors (*Figure 2B-C* and *Figure 2—figure supplement 1B-C*). If the fluorescence signal ($\Delta R(t)/R_0$) reflects a change of intracellular free calcium concentration [$Ca^{2+}$], the ramping rate, defined as $\zeta = \frac{dR}{dt}$ (*Figure 2C*), would be proportional to the calcium current. Higher $\zeta$ may reflect a larger depolarization of the neuronal membrane potential. In *Figure 2B-C*, 76% trials (91/120) in the type-I (RF) transition show a positive ramping rate, whereas the proportion rose to 95% (109/115, $p < 0.0001$, $\chi^2$ test) in the type-II (RT) transition. Among trials longer than 1.5 seconds, they all showed positive $\zeta$, which during the type-II transition was significantly higher than that during the type-I transition (*Figure 2C* and *Figure 2—figure supplement 1C*). These results suggest that the more active AIB are, the more likely a worm would terminate its reversal with a turn.

Optogenetic activation of AIB (P*npr-9*::ChR2 or P*npr-9*::Chrimson) alone reliably triggered reversals followed by omega turns (*Figure 2D*, *Figure 2—figure supplement 1D* and *Figure 2—video 1*), whereas strong optogenetic inhibition of AIB (P*mec-4*::ChR2; P*npr-9*::Arch; *lite-1*) during ALM/AVM induced escape responses almost completely abolished omega turns (*Figure 2D*). We also generated transgenic animals in which AIB were persistently hyperpolarized by an expression of exogenous potassium channels (P*npr-9*::TWK-18(*gf*)). Interestingly, the no-response fraction increased to ~20% (70/332, p<0.0001, $\chi^2$ test) upon stimulating ALM/AVM in these animals and a significantly larger portion of responses were pauses (35/332, p<0.0001, $\chi^2$ test, *Figure 2E*). We did not observe a significant change in the type-II transition rate $r_2$ (Kolmogorov-Smirnov test, p=0.6, *Figure 2F*), which might be due to a weaker AIB inhibition in these animals. Furthermore, optogenetic ablation of AIB alone (P*npr-9*::PH-miniSOG, *Figure 2—figure supplement 1H*; *Qi et al., 2012*; *Xu and Chisholm, 2016*) significantly suppressed the type-II transition rate (Kolmogorov-Smirnov test, p<1e-20, *Figure 2—figure supplement 1I*). RIM (*Figure 2A*) also promoted reversals (*Figure 2—figure supplement 1B and E–G*), but were less important in modulating the type-II transition (*Figure 2—figure supplement 1D–G*). Together, our data strongly suggest that AIB play an important role in promoting reversal and turning behaviors.

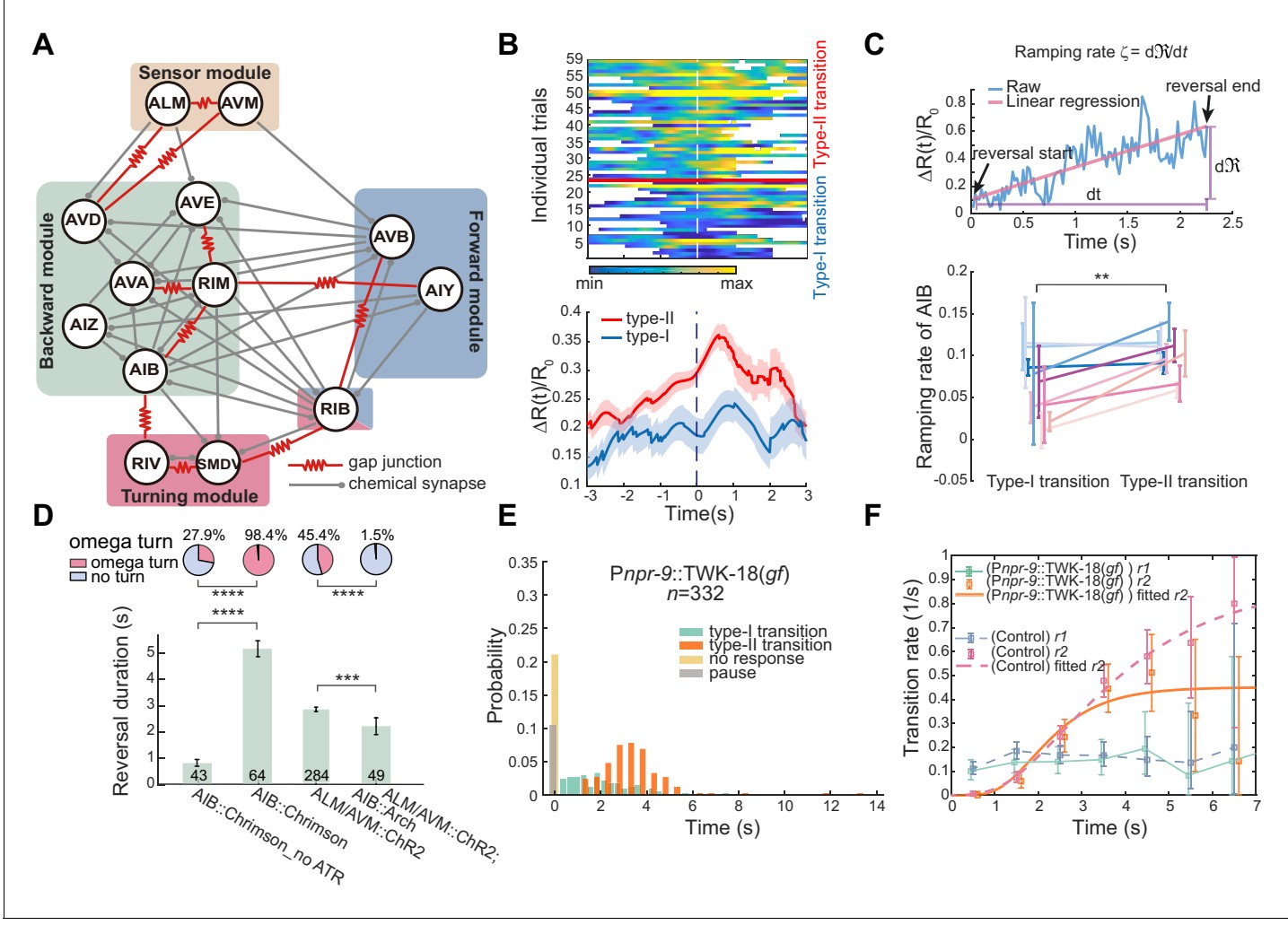

**Figure 2.** Interneurons AIB are crucial for motor state transitions. (A) Simplified circuit diagram underlying the gentle touch induced escape responses. Neurons were grouped into four modules based on their functional roles and activity patterns. (B) Calcium activity of AIB during spontaneous reversals before type-I ($n = 23$) and type-II ($n = 36$) transitions in unrestrained behaving animals (P$inx$-$1$::GCaMP6;P$inx$-$1$::wCherry). Here, data are aligned to the ends of reversals (vertical dashed line, $t = 0$). Heat map across trials (up) and $\Delta R(t)/R_0$ (Mean ± SEM, bottom) are shown. (C) Ramping rate of calcium activity in AIB. Up, raw single trial $\Delta R(t)/R_0$ from reversal start to reversal end. The ramping rate is the slope of the red line, fitted by linear regression. Bottom, ramping rates of AIB during type-I and type-II transitions. Each color (Mean ± SEM) represents single animal data across multiple trials. Total nine animals (P$inx$-$1$::GCaMP6;P$inx$-$1$::wCherry) were tested. Very short reversals (less than 1.5 s) are excluded, for some of them have negative ramping rates and the slope estimate is susceptible to noise (but including those trials doesn't affect our conclusion). **p<0.01, two-way ANOVA. (D) Optogenetic activation of AIB (635 nm, 4.46 mW/mm², 7 s) or inhibition of AIB (561 nm, 21.71 mW/mm², 12 s) during ALM/AVM (473 nm, 14.71 mW/ mm², 1.5 s) triggered avoidance behaviors, reversal durations (bar graph) and fractions of animals executing omega turns (pie chart) are shown. Error bars are SEMs. Bar graph, Mann–Whitney U test. Pie chart, $\chi^2$ test. *p<0.05, ***p<0.001, ****p<0.0001. Here and below, the actual turning percentages ($n_{turn}/n_{total}$) are noted beside the pie chart and numbers within the bars indicate the number of trials with reversal. (E–F) Reversal length distribution (E) and transition rates (F) during escape responses when AIB were persistently hyperpolarized through an exogenous expression of the potassium channel TWK-18. Control group is from *Figure 1C*.

The online version of this article includes the following video, source data, and figure supplement(s) for figure 2:

**Source data 1.** Source data for *Figure 2* and *Figure 2—figure supplement 1*.

**Figure supplement 1.** Local interneurons AIB and RIM in the backward module differentially modulate motor state transitions.

**Figure 2—video 1.** AIB neurons are crucial for turn and backward movement, Related to *Figure 2* and *Figure 2—figure supplement 1*.

https://elifesciences.org/articles/56942#fig2video1

## Feedforward coupling between the backward module and the turning module drives the omega turn

How do AIB drive turning behaviors? Whole brain imaging in immobilized animals implicated that AIB and their electrically-coupled partners RIV (*Figure 2A* and *Figure 3—figure supplement 1*) exhibited sequentially activated patterns (*Kato et al., 2015*). We compared RIV activity patterns (P*lim-4*::GCaMP6) underlying different motor sequences during spontaneous behaviors. During the type-II (RT) transition, RIV calcium signal rose rapidly immediately before a turn began, whereas it remained largely quiescent during the type-I (RF) transition (*Figure 3A* and *Figure 3—figure supplement 2A*). The calcium signal decayed towards baseline before the animal finished the turn and resumed forward movement (*Figure 3—figure supplement 2B*).

To directly probe the functional connectivity between AIB and RIV, we performed simultaneous optogenetic stimulation of AIB (P*npr-9*::Chrimson) and calcium imaging of RIV (P*lim-4*::GCaMP6:: wCherry, *Figure 3B*). In immobilized wild-type animals, upon stimulating AIB at $t = 0$, RIV calcium signal rapidly rose (*Figure 3C* dark blue and *Figure 3—figure supplement 2C*). Several innexin, including INX-1, UNC-7 and UNC-9, have been reported to be expressed in AIB and RIV (*Altun et al., 2009*; *Bhattacharya et al., 2019*). Some of these innexins were shown to form homotypic and/or heterotypic gap junctions (*Kawano et al., 2011*; *Liu et al., 2013*; *Starich et al., 2009*; *Xu et al., 2018*). To determine whether electrical synapses contribute to the observed functional coupling between AIB and RIV, we examined the effect of AIB stimulation in *inx-1unc-9unc-7* triple innexin mutants. RIV remained quiescent upon AIB stimulation (*Figure 3C* red and *Figure 3—figure supplement 2D*), indicating that gap junction coupling underlies AIB stimulation-mediated RIV calcium activity.

UNC-7 and UNC-9 are broadly expressed in the motor circuit, and *unc-7* or *unc-9* mutants exhibit uncoordinated movements that prohibit them from completing a motor sequence (*Barnes and Hekimi, 1997*; *Brenner, 1974*; *Kawano et al., 2011*; *Starich et al., 1993*; *Xu et al., 2018*). *inx-1* single mutants exhibit superficially normal forward and backward movements, allowing us to examine the behavioral requirement of INX-1. The presence of multiple innexins in many *C. elegans* neurons implicates that they may function redundantly at electrical synapses. Consistent with this notion, we find that optogenetic activation of AIB was capable, but with less likelihood, to trigger a turn in *inx-1* mutants (*Figure 3D*). Rescuing *inx-1* in AIB was sufficient to restore the turning probability (*Figure 3D*). Because *inx-1* mutants were still capable of generating omega turns, we propose that either multiple innexins between AIB and RIV, or parallel circuit pathways are at play.

When we performed dual optogenetic activation and calcium imaging in wild-type animals that were allowed to move, an increase of RIV calcium activity was also observed. But we observed a delay in RIV calcium signal, with its increase arriving at variable times (*Figure 3E* and *Figure 3—figure supplement 2F*) instead of an immediately rise after stimulation onset. In many trials, the rise of RIV calcium activity coincided with the initiation of an omega turn (*Figure 3—figure supplement 2F*), an event that was used to realign all trials at $t = 0$ (*Figure 3E* and *Figure 3—figure supplement 2F*).

Delayed depolarization of RIV in a moving animal may result from a convergence of excitatory- and inhibitory- inputs onto the turning module (*Figure 2A*). When neural activity in behaving animals was aligned to the onset of optogenetic stimulation, a transient quiescence or decrease of RIV calcium activity indeed appeared after $t = 0$ (*Figure 3F* and *Figure 3—figure supplement 2F*). We hypothesized that a rapid increase of calcium activity in *Figure 3C* (dark blue) could result from a stronger depolarization of RIV neurons in immobilized animals. Consistently, when the calcium imaging experiment in immobilized animals was combined with a weak and persistent optogenetic inhibition of RIV (P*lim-4*::Arch), we also observed a delayed and rectified excitation in RIV (*Figure 3C* light blue and *Figure 3—figure supplement 2E*).

Taken together, our data suggest that the feedforward excitation between the backward module and the turning module takes the form of electrical synapses, likely between AIB and RIV. We considered an effective functional coupling through polysynaptic excitation highly unlikely. First, AIB triggered ventral-biased turning behaviors did not require glutamatergic synaptic transmission (*Figure 4A*), thus excluding polysynaptic pathways via chemical synapses from AIB. Second, two possible polysynaptic excitation pathways from AIB via electrical coupling involve interneurons RIM or RIS (*Figure 3—figure supplement 1*), revealed by the *C. elegans*

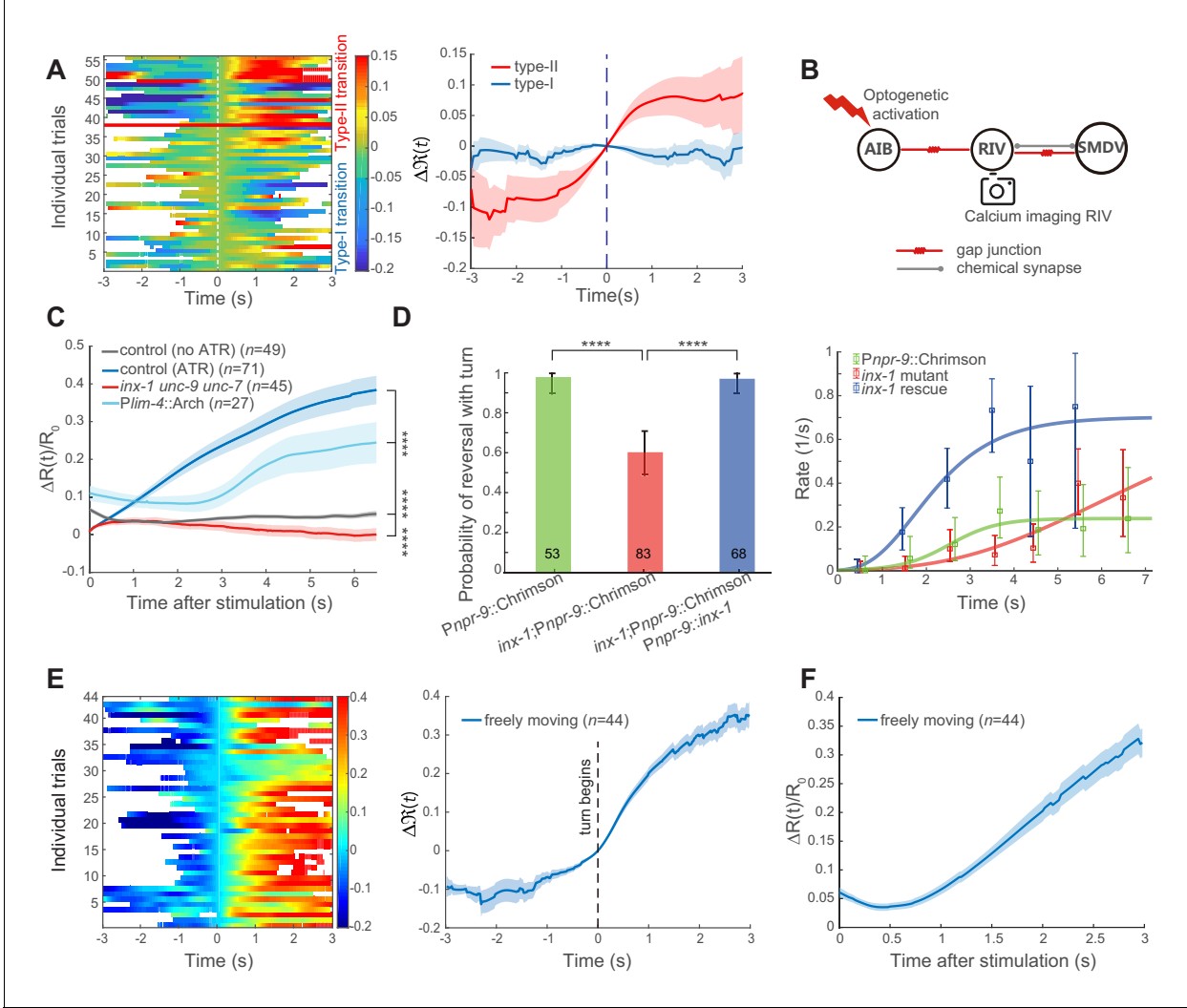

**Figure 3.** Feedforward excitation via electrical coupling drives omega turns. (**A**) Calcium activity of RIV during the type-I (*n* = 38) and the type-II (*n* = 18) transitions in spontaneous behaving animals (P*lim-4*::GCaMP6). Here, data are aligned to the ends of reversals (vertical dashed line, *t* = 0). Heat map across trials (left) and $\Delta\mathfrak{R}(t)$ (Mean ± SEM, right, also see Materials and Methods) are shown. (**B**) Optical neurophysiology for probing the feedforward coupling between the backward module and the turning module. (**C**) Simultaneous optogenetic activation of AIB (635 nm, 6.11 mW/mm$^2$) and calcium imaging of RIV in immobilized animals. $\Delta R(t)/R_0$ (Mean ± SEM) under different genetic backgrounds are shown: control (ATR) is wild-type (dark blue); *inx-1unc-9unc-7* triple mutant (red); calcium imaging of RIV in the presence of moderate inhibition (561 nm, 1.94 mW/mm$^2$) of RIV (light blue, P*lim-4*:: Arch). The control (no ATR) represents imaging data from wild-type animals without feeding all-trans retinal (grey). ****p<0.0001, two-way ANOVA. (**D**) Probability of a reversal followed by a turn (left) and type-II transition rates (right) for gap junction deficient mutants. AIB, expressing Chrimson, were optogenetically activated for 7 s (635 nm, 4.46 mW/mm$^2$) in wild-type animals (green), gap junction deficient mutants *inx-1* (red), and *inx-1* mutants, in which INX-1 channels were restored specifically in AIB (blue). Error bars indicate 95% binomial proportion confidence interval. $\chi^2$ test. ****p<0.0001. (**E**) Simultaneous optogenetic activation of AIB (635 nm, 6.11 mW/mm$^2$) and calcium imaging of RIV in unrestrained behaving animals. Left, calcium activity heatmap across trials. *t* < 0 represents reversals. Omega turns start at *t* = 0. Right, $\Delta\mathfrak{R}(t)$ (mean ± SEM) are shown (blue). (**F**) Data are related to (E), but *t* = 0 is aligned to the beginning of AIB stimulation.

The online version of this article includes the following video, source data, and figure supplement(s) for figure 3:

**Source data 1.** Source data for *Figure 3*.

**Figure supplement 1.** Possible polysynaptic pathways from AIB to RIV (less than four layers are shown) according to *C. elegans* connectome.

**Figure supplement 2.** Optogenetic activation of AIB and calcium imaging of RIV in different experimental conditions.

**Figure 3—video 1.** Optogenetic activation of AIB under various genetic backgrounds, Related to *Figure 3* and *Figure 4*.

https://elifesciences.org/articles/56942#fig3video1

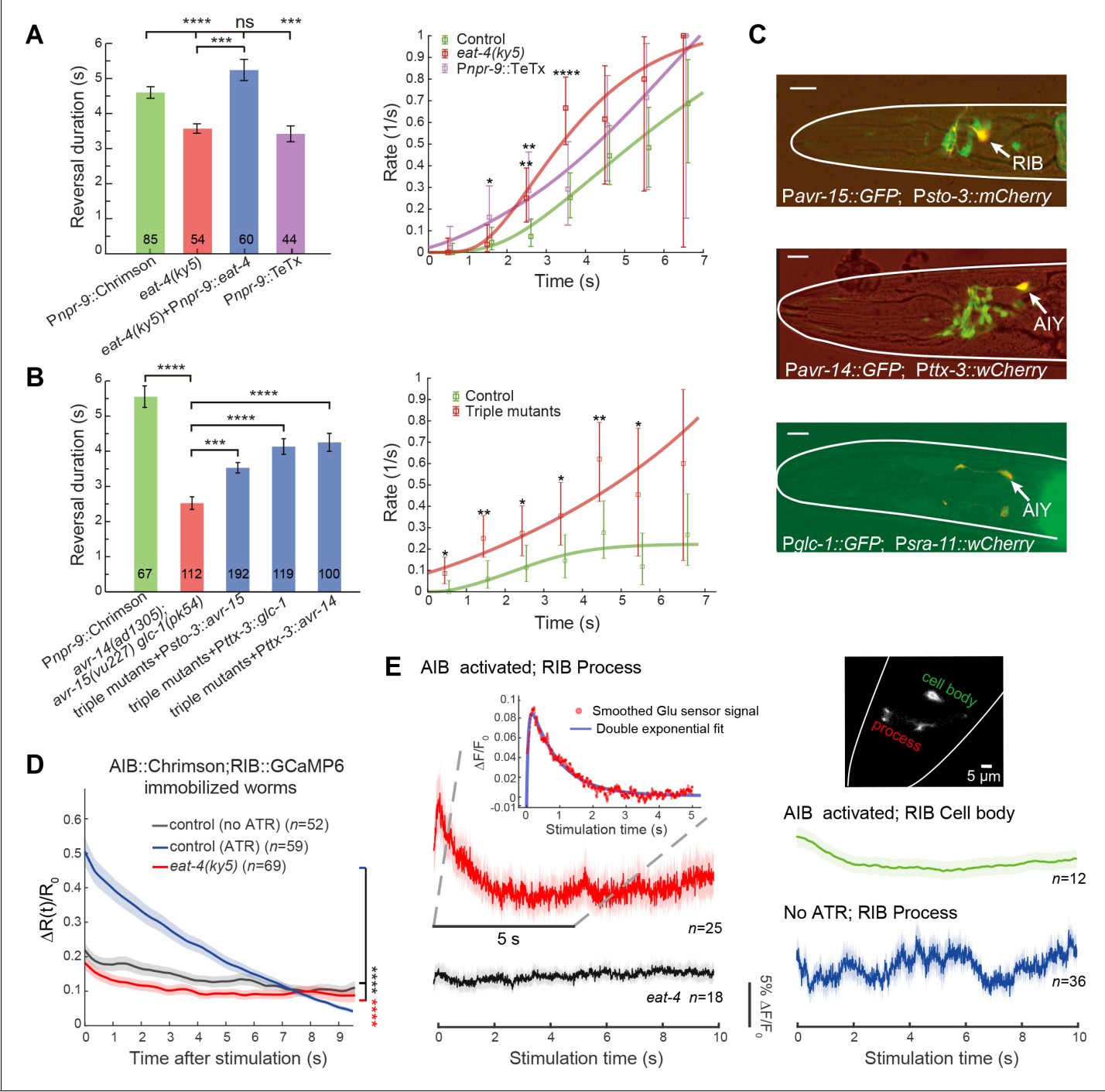

**Figure 4.** Glutamatergic inhibition modulates type-II transition. (A) Reversal durations (left) and type-II transition rates (right) in glutamatergic synaptic transmission deficient animals upon strong and persistent optogenetic activation of AIB. Optogenetic stimulation was delivered for 7 s using red light (635 nm, 4.46 mW/mm$^2$). Right, compare $r_2$ in *eat-4* mutant or in P*npr-9*::TeTx with control animals across the whole distribution (Kolmogorov-Smirnov test, *p*=6.2e-7, *p*=0.0026) or within a time window ($\chi^2$ test: * *p*<0.05, ** *p*<0.01, *** *p*<0.001, **** *p*<0.0001). (B) Reversal durations (left) and type-II transition rates (right) in GluCl receptor mutants upon optogenetic activation of AIB (635 nm, 4.46 mW/mm$^2$). Right, compare $r_2$ in triple receptor mutant with control animals across the whole distribution (Kolmogorov-Smirnov test, *p*=5.9e-8) or within a time window ($\chi^2$ test: * *p*<0.05, ** *p*<0.01). (C) Genes encoding GluCl receptors were expressed in local interneurons RIB and AIY respectively. GFP reporter lines were constructed using *avr-14*, *avr-15* and *glc-1* promoters, respectively; mCherry (or wCherry) reporters were used for cell identification. Expression pattern from one section is showed. Scale bar, 10 μm. (D) Simultaneous optogenetic activation of AIB (635nm, 6.11 mW/mm$^2$) and calcium imaging of RIB in immobilized animals under wide-type (control (ATR)) (blue) or *eat-4* glutamate deficient mutant background (red). $\Delta R(t)/R_0$ (Mean ± SEM) are shown. *t* = 0 represents the

*Figure 4 continued on next page*

*Figure 4 continued*

beginning of AIB stimulation. The control group (no ATR) (grey) represents imaging data from animals without feeding all-trans retinal. ****$p<0.0001$, two-way ANOVA. (E) Functional coupling between AIB and RIB neurons was directly tested through glutamate imaging. Upon stimulation of AIB (P*npr-9*::Chrimson), the process (red), cell body (green) of the RIB (P*sto-3*::iGluSnFR) and the RIB process in *eat-4* glutamate deficient mutants (grey) exhibited distinct fluorescence signals. The control group (blue) represents imaging data from animals without feeding all-trans retinal. Raw iGluSnFR imaging was recorded at 150 Hz. Trial average (bold color) and SEM (shaded region) are shown. Inset, the fluorescence signal of the process (red) is fitted with a double exponential function (blue), $B\left(e^{-\frac{t}{\tau_1}} - e^{-\frac{t}{\tau_2}}\right) + c$, where $B$, $\tau_1, \tau_2$ and $c$ are free parameters. $\tau_1$ represents the time constant of glutamate signal decay, and $c$ is the baseline constant, which was subtracted in the inset plot. For reversal duration, Error bars indicate SEM. Mann–Whitney U test: ***$p<0.001$, ****$p<0.0001$. All multiple comparisons were adjusted using Bonferroni correction.

The online version of this article includes the following source data and figure supplement(s) for figure 4:

**Source data 1.** Source data for *Figure 4* and *Figure 4—figure supplements 1–2*.
**Figure supplement 1.** Glutamatergic inhibitions between AIB and RIB flexibly control the motor state transitions.
**Figure supplement 2.** Functional coupling between AIB and RIB neurons was directly tested through glutamate imaging.

connectome (*White et al., 1986*). Our optogenetic activation of AIB while inhibiting RIM did not modify the turning probability, whereas activating RIM while inhibiting AIB significantly reduced the turning probability (*Figure 2—figure supplement 1F–G*). Activation of RIS would drive an animal to a pause state and abolish motor actions (*Steuer Costa et al., 2019*). Both results argue against RIM and RIS being directly involved in driving turning behaviors.

## Inhibitory glutamatergic synaptic transmission modulates the type-II transition

We next investigated behavioral flexibility in the timing of an action. Given the feedforward coupling between the backward module and the turning module, we asked why omega turns did not immediately follow the optogenetic activation of AIB. We hypothesized that a balance of feedforward excitation and an unknown inhibition provides a potential mechanism to shape the statistics of the type-II (RT) transition. Besides gap junctions, AIB make chemical synapses with neurons in other modules (*Figure 2A* and *Figure 4—figure supplement 1A*), which may facilitate this inhibition. AIB are glutamatergic (*Serrano-Saiz et al., 2013*). In the glutamate vesicular transport deficient mutant *eat-4*, optogenetic activation of AIB robustly induced an omega turn that was preceded by a much shorter reversal (*Figure 4A* left and *Figure 4—figure supplement 1B*). Restoring *eat-4* expression specifically in AIB significantly prolonged the reversal length before the onset of an omega turn (*Figure 4A* left, *Figure 4—figure supplement 1B* and *Figure 3—video 1*). Likewise, stimulating AIB while blocking chemical synaptic transmission from AIB (P*npr-9*::TeTx) triggered an omega turn preceded by a shorter reversal (*Figure 4A* left and *Figure 3—video 1*). Consistently, $r_2$ rose more rapidly when glutamatergic inputs from AIB were disrupted (*Figure 4A* right).

*C. elegans* nervous system possesses a family of inhibitory glutamate-gated chloride (GluCl) channels (*Dent et al., 1997*). Upon optogenetic stimulation AIB, the triple GluCl mutant *avr-14*(*ad1035*); *avr-15*(*vu227*)*glc-1*(*pk54*) exhibited a behavioral phenotype resembling that of the *eat-4* mutant (*Figure 4B* and *Figure 4—figure supplement 1B*). In some trials (19/112, $p<0.0001$, Fisher's exact test), AIB stimulation immediately triggered omega turns without delay (*Figure 3—video 1*). This suggests that postsynaptic GluCl receptors work synergistically in modulating the onset timing of a turn. Using GFP reporter lines, we found *avr-14*, *glc-1* and *avr-15* expressed in many neurons. By focusing on overlaps with neurons known to encode motor states (*Figure 2A*), we found that P*avr-14* and P*glc-1* reporters exhibited expression in AIY interneurons, while the P*avr-15* reporter exhibited expression in RIB interneurons (*Figure 4C*).

Unlike AIY, RIB receive more and invariant synaptic inputs from AIB (*White et al., 1986*; *Witvliet et al., 2020*; *Figure 2A*), and hence are the prominent postsynaptic partners of AIB. Consistent with a glutamate mediated feedforward inhibition, RIB calcium activity (P*sto-3*::GCaMP6) significantly reduced upon optogenetic activation of AIB in immobilized animals, which was not observed in the glutamate vesicular transport deficient mutant *eat-4* animals (*Figure 4D* and *Figure 4—figure supplement 1C*). Moreover, restoring either *avr-15* expression in RIB or *avr-14* (or *glc-1*) expression in AIY promoted a longer reversal before the onset of an omega turn (*Figure 4B* and *Figure 3—video 1*) upon AIB activation.

To further investigate the functional connectivity between AIB and RIB, we imaged glutamate signaling (*Marvin et al., 2013*) at RIB (P*sto-3*::iGluSnFR) upon persistent optogenetic stimulation of AIB (P*npr-9*::Chrimson; see Materials and methods). After the onset of stimulation, a rapid rise (*Figure 4E* inset plot,~10%$\Delta F/F_0$) of iGluSnFR signal on RIB's neurites (*Figure 4E* and *Figure 4—figure supplement 2A* up) was followed by a slow decay. The fluorescence signal change was well fit by

$$\frac{\Delta F(t)}{F_0} = B\left(e^{-\frac{t}{\tau_1}} - e^{-\frac{t}{\tau_2}}\right),\tag{1}$$

where $\tau_1 = 0.9\,s$, and $\tau_2 = 100\,ms$. In animals without feeding all-trans retinal (a co-factor required for AIB optogenetic stimulation), we observed random and smaller amplitude (~5%$\Delta F/F_0$) fluctuations of iGluSnFR signals (*Figure 4E*). Such dynamics was not observed in the glutamate vesicular transport deficient mutant *eat-4* animals (*Figure 4E* and *Figure 4—figure supplement 2A* bottom).

## Local interneurons RIB promote both turning and forward behaviors

Local interneurons RIB, together with the ventral cord-projecting premotor interneurons AVB, have been previously reported to encode forward movement state (*Gray et al., 2005*; *Kato et al., 2015*; *Li et al., 2014*). Moreover, RIB form gap junctions with SMDV (*Figure 2A*), motor neurons that have also been implicated in ventral biased omega turns (*Gray et al., 2005*; *White et al., 1986*). RIB calcium activity declined during reversals, and rose during the type-I (RF) and type-II (RT) transitions (*Figure 5A* and *Figure 5—figure supplement 1A*), which is characteristic of neurons in both forward and turning modules (*Figure 2A*).

Optogenetic manipulation of RIB in freely behaving animals further revealed their functions during motor control. Activating RIB (P*sto-3*::Chrimson) during reversals triggered a transition to either an omega turn or forward movement (*Figure 5B* left). Strong optogenetic activation (635 nm, 3.75 mW/mm$^2$) of RIB during forward movement reliably triggered omega turns (*Figure 5B* right and *Figure 5—video 1*). On the other hand, inhibiting RIB (P*sto-3*::Arch) during forward movement led to a pause state (*Figure 5B* right and *Figure 5—video 1*).

When RIB interneurons were directly inhibited to mimic an inhibitory synaptic input, either optogenetically or by an expression of histamine-gated chloride channels (*Pokala et al., 2014*), the type-II (RT) transition rate $r_2$ plateaued at a significantly reduced value (*Figure 5C* and *Figure 5—figure supplement 1B*). Consistently, an escape response comprised of a much longer reversal before an omega turn was initiated (*Figure 5C*, *Figure 5—figure supplement 1B* and *Figure 5—video 2*), in agreement with the GluCl rescue results (*Figure 4B* left). The type-I transitions were also largely suppressed (*Figure 5C*), as RIB also potentiate forward movement. Optogenetic ablation (P*sto-3*::miniSOG) or blocking chemical synaptic transmission (P*sto-3*::TeTx) from RIB also led to prolonged reversals during ALM/AVM-triggered escape responses (*Figure 5F*).

We asked how RIB may mediate neural activity in the turning module. Upon optogenetic stimulation of AIB (*Figure 5D–E*), the rise of RIV calcium activity in RIB ablated animals showed the same rectified activation when all trials were aligned to the beginning of a turn (*Figure 5E*). However, RIV activity was preceded by a longer quiescent state when trials were aligned to the stimulus onset (*Figure 5D* and *Figure 5—figure supplement 1C*). Thus, RIB modulate motor state transitions in part through indirect modulation of the timing of RIV activation (*Figure 2A*).

## Inhibitory feedback contributes to reversal termination

The beginning of a turn marks the end of a reversal. We next asked whether the type-II (RT) transition can be accounted for by self-termination of neural activity in the backward module (*Figure 2A*), analogous to a feedforward synaptic chain model, or, whether activation of the turning module provides a feedback inhibition to terminate the activity in the backward module.

In a feedforward synaptic chain model, perturbing neural activity in the downstream neurons would not affect the dynamics of upstream neurons. To test this model, we generated transgenic animals that express Archaerhodopsin in RIV/SAA/SMB neurons (P*lim-4*::Arch; *Figure 6A*). RIV/SAAD all exhibited elevated calcium activity during omega turns, and could be regarded as downstream outputs of the backward module (*Figure 5—figure supplement 1D*). However, optogenetic

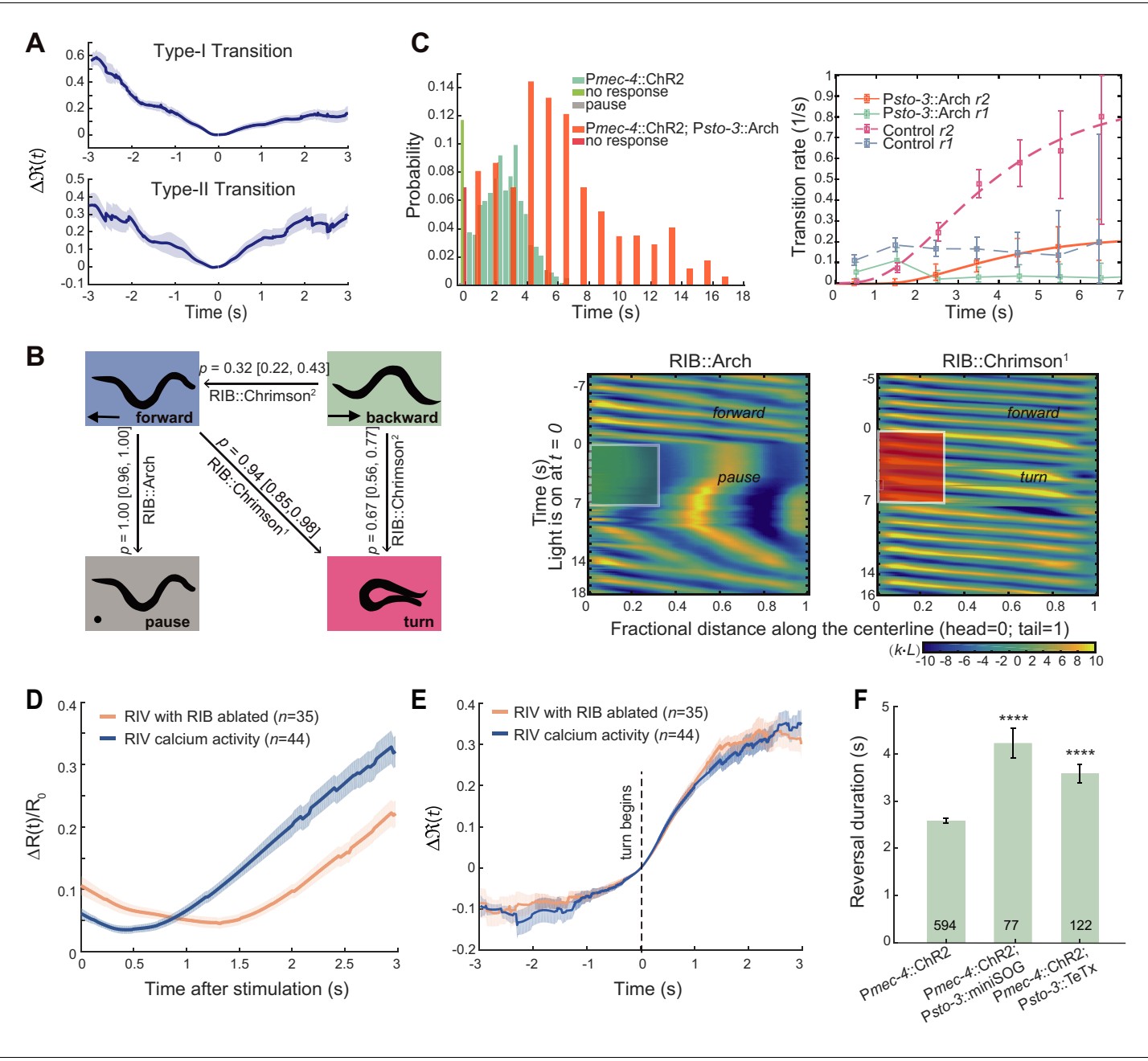

**Figure 5.** Local interneurons RIB promote both turning and forward behaviors. (**A**) Calcium activity of RIB during type-I (up, *n* = 41) and type-II transitions (bottom, *n* = 25). Δ𝕽(*t*) (mean ± SEM) are shown. *t* = 0 is aligned to the initiation of forward movement (up) or an omega turn (bottom). (**B**) Statistics of motor state transitions (left) and representative curvature kymographs (right) upon optogenetic manipulation of RIB. Left, the probability for a transition and its 95% confidence limits were computed. RIB::Chrimson[1], *n* = 65, red light (635 nm, 3.75 mW/mm²); RIB::Chrimson[2], *n* = 82, red light (635 nm, 1.00 mW/mm²); RIB::Arch, *n* = 83, green light (561 nm, 8.14 mW/mm²). Right, animals crawled on fresh agar plates. Body curvature was normalized by a *k* · *L*, where *L* is the body length. Green (or red) shaded regions show selected spatiotemporal regions for optogenetic inhibition (or activation). The kymograph of turning behaviors exhibits longer cycles to complete body bending and larger body curvature, which are different from those during forward movement. (**C**) Reversal length distributions (left) and transition rates (right) when ALM/AVM activation was followed by optogenetic inhibition of RIB (12 s green light, 561 nm, 1.94 mW/mm²). P*mec-4*::ChR2;P*sto-3*::Arch, *n* = 173. Control group is from ***Figure 1C***. (**D**) Calcium imaging of RIV in wild-type (blue) and RIB-ablated animals (orange) upon optogenetic stimulation of AIB in freely behaving animals. Δ*R*(*t*)/*R*₀ (Mean ± SEM) are shown. (**E**) Data are related to (**D**), but all trials were aligned to the onset of omega turns. (**F**) Reversal length during ALM/AVM-triggered escape responses in RIB-ablated animals or in animals where chemical synaptic transmission from RIB was blocked. Error bars indicate SEM. Mann–Whitney U test, ****p<0.0001. All multiple comparisons were adjusted using Bonferroni correction.

The online version of this article includes the following video, source data, and figure supplement(s) for figure 5:

*Figure 5 continued on next page*

*Figure 5 continued*

**Source data 1.** Source data for *Figure 5* and *Figure 5—figure supplement 1*.

**Figure supplement 1.** RIB receive feedforward inhibition to modulate type-II transition and neurons in the turning module might contribute to reversal termination.

**Figure 5—video 1.** Optogenetic manipulation of RIB can induce motor state transitions, Related to *Figure 5*.

https://elifesciences.org/articles/56942#fig5video1

**Figure 5—video 2.** Optogenetic inhibition of RIB or RIV/SAA/SMB neurons during escape responses prolonged reversal durations, Related to *Figure 5* and *Figure 6*.

https://elifesciences.org/articles/56942#fig5video2

inhibition of RIV/SAA/SMB or RIV alone by spatially patterned illumination during escape responses promoted significantly longer reversals (*Figure 6B* and *Figure 5—video 2*).

Observations from optogenetic ablation of RIV/SAA/SMB (P*lim-4*::miniSOG) also argue against a pure feedforward synaptic chain model. The type-II transition was abolished since animals could no longer generate a complete omega turn (*Figure 6C* upper panel), while the ability of direct transition from a backward to a forward movement remained unaffected and the type-I (RF) transition rate $r_1$ remained similar (*Figure 6C* upper panel) to wild-type animals. Notably, the reversal duration became much longer and approached 30 s in some trials, which had not been observed in wild-type animals (*Figures 1C* and *6C* upper panel and *Figure 6—video 1*). These results indicate that during normal type-II transitions, persistent neural activity in the upstream backward module could be abolished through inhibitory feedback from the downstream activity in the turning module.

Both the type-I (RF) transition rate (*Figure 6C* upper panel) and the mirror transition rate (FR) from a forward movement to a spontaneous reversal in wild-type animals (*Figure 6C* bottom panel) are consistent with the homogeneous Poisson process at long timescale, leading to exponential survival functions (*Figure 6C* insets) — fraction of backward or forward movements survived to $t$ (*Berg, 1993*; *Stephens et al., 2011*). We did not observe an exponential survival function of reversals in wild-type animals. In the *absence* of the turning module, the statistics of forward and backward movements (*Figure 6C*) became consistent with a simple dynamic model, where a system stochastically transitions between two attractor states at constant rates.

Together, our data suggest that the feedforward inhibition (*Figure 4*) and feedback inhibition (*Figure 6*) between the backward module and the turning module implement a winner-take-all computation for action selection. The motor module with the highest level of activity stays active by suppressing the activities of other modules.

## A biophysical model of the type-II transition

With both structural and functional evidence, we now propose a mathematical model for the type-II (RT) transition. The turning module, represented by RIV inter/motor neurons, receives opposing excitatory and inhibitory inputs during a reversal (*Figure 7A*). The rapid increase of RIV activity coincides with the beginning of an omega turn (*Figure 3A and E*). To capture the essential process, we assumed that the membrane potential of RIV $x$ fluctuates around a balanced state $x_0$ during a reversal (*Figure 7B*), and its neural dynamics is governed by the Langevin equation:

$$\frac{dx}{dt} = -k(x - x_0) + \eta, \quad x < x_{th} \tag{2}$$

where $k$ depends on, among others, the gap junction and inhibitory synaptic conductances (see Appendix); and $\eta$ could be regarded as fluctuations in synaptic currents (*Lindsay et al., 2011*; *Narayan et al., 2011*) and other sources of noises that are not explicitly considered in the model. For simplicity, $\eta$ is treated as uncorrelated Gaussian white noise: $\langle \eta(t)\eta(t') \rangle = 2\sigma^2\delta(t - t')$. Once the membrane potential crosses the threshold $x_{th}$, RIV become rapidly depolarized due to a nonlinear rectified activation (*Figure 7B*), immediately terminating the reversal via feedback inhibition and starting a turn by activating ventral muscles (*Figure 7A and D*).

The next step is to calculate the type-II transition rate $r_2$: the probability that $x$ crosses $x_{th}$ per unit time. It is currently impossible to measure $k$, but we can proceed by making a prediction. Based on

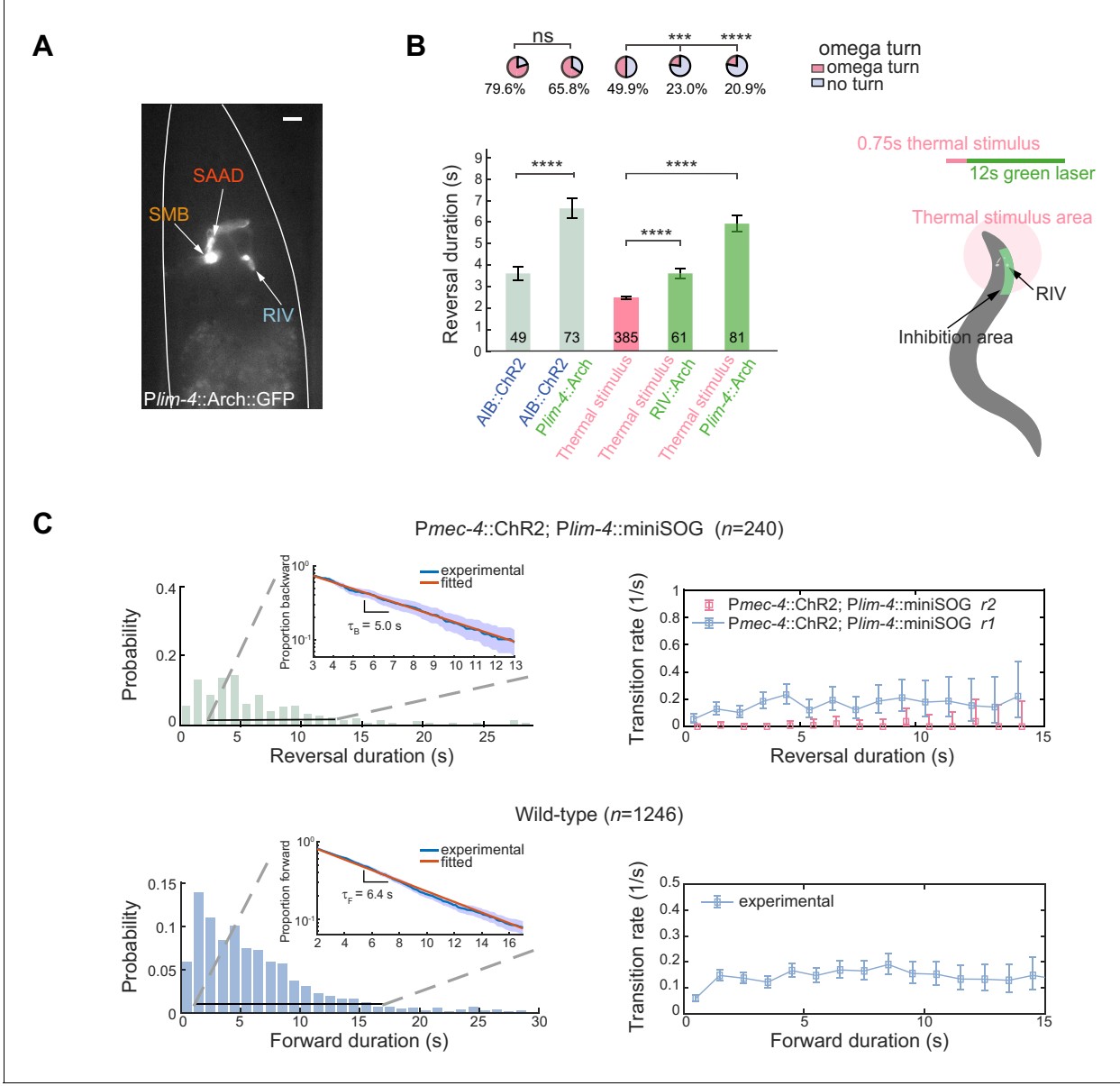

**Figure 6.** Feedback inhibition from the turning module contributes to reversal termination. (A) Expression pattern of P*lim-4*::Arch::GFP. Scale bar, 10 μm. (B) Left, escape responses induced by optogenetic activation of AIB (blue text, 473 nm, 14.71 mW/mm², 1.5 s) or thermal stimulus (pink text, 1480 nm, 0.75 s) followed by optogenetic inhibition of Arch-expressed interneurons (green text, 561 nm, 21.71 mW/mm², 12 s). Bar graph, reversal durations, Mann–Whitney U test and error bars are SEMs. Pie chart, fractions of trials executing omega turns, χ² test. *p<0.05, ***p<0.001, ****p<0.0001. All multiple comparisons were adjusted using Bonferroni correction. Right, schematic diagram for selective inhibition of RIV near animal head during thermally induced (1480 nm, 0.75 s) escape responses. (C) Up, reversal length distribution (left) and transition rate (right) in RIV/SAAD/SMB ablated animals. Bottom, spontaneous run length distribution (left) and transition rate from forward to backward movements (right) in wild-type animals. Insets are plots of survival functions: proportion of trials moving backward (up) or forward (bottom) until *t*. We fitted only trials with reversal duration ≥3 s and forward duration ≥2 s.

The online version of this article includes the following video and source data for figure 6:

**Source data 1.** Source data for *Figure 6*.

**Figure 6—video 1.** RIV neurons are essential for omega turns, Related to *Figure 6*.

https://elifesciences.org/articles/56942#fig6video1

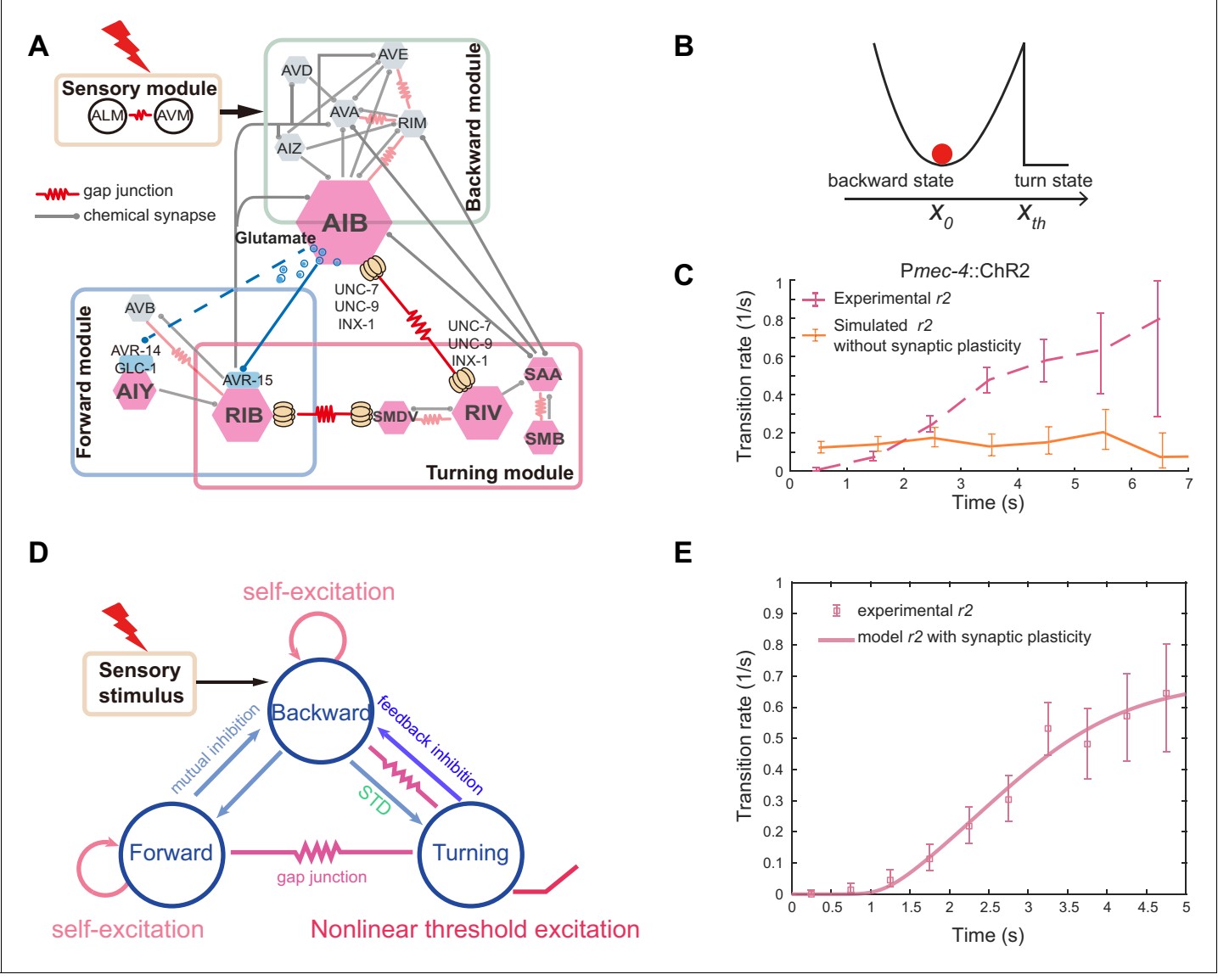

**Figure 7.** A model summary for motor sequence generation. (**A**) Schematics of neuronal circuit of *C. elegans* avoidance behavior. Feedforward excitation between local interneurons AIB and RIV triggers turning behavior. Glutamatergic inhibition between AIB and RIB flexibly controls the motor state transitions. Feedback inhibition from the turning module helps terminate backward behavior. (**B**) Illustration of the biophysical meaning of the type-II transition. The sub-threshold membrane potential of RIV fluctuates around the fixed point $x_0$, just like a particle (red) in an energy well. When RIV membrane potential crosses the threshold $x_{th}$, RIV along with other neurons in the turning module become fully activated and a turn starts. (**C**) A simple stochastic model without short-term synaptic plasticity cannot account for the experimentally observed transition rate. (**D**) Schematics of a three-module model based on the animal connectome and experimentally identified functional motifs. Short-term synaptic depression (STD) was introduced in the feedforward inhibition. (**E**) Type-II transition rate during ALM/AVM triggered escape responses. Pink line is a theoretical fit using *Equation 4*. $\tau_g$ in *Equation 4* is given by the glutamate decay constant in *Figure 4E*.

electrophysiological recordings of *C. elegans* interneurons (*Lindsay et al., 2011*; *Roberts et al., 2016*), the membrane time constant of a neuron (~10 milliseconds) is much smaller than the behavioral timescale (~seconds). As a result, the membrane potential of a model neuron rapidly approaches the fixed point $x_0$. By solving this problem analytically using one-dimensional Fokker-Planck equation near the system equilibrium (see Appendix), we find

$$r_2 \approx \frac{k}{\pi} erfi^{-1}\left[\sqrt{\frac{k}{2\sigma^2}}(x_{th} - x_0)\right] \tag{3}$$

Here $erfi(x) = \frac{2}{\sqrt{\pi}}\int_0^x e^{z^2}\,dz$ is the imaginary error function.

*Equation 3*, however, would lead to a constant rate on the behavioral timescale, like that during the type-I (RF) transition, as expected and confirmed by our computer simulation (*Figure 7C*). To explain the experimental observation of the rising phase of $r_2$ (*Figure 1C–D*), we incorporated a plasticity mechanism analogous to short-term synaptic depression (STD): the feedforward inhibition from the backward module to the turning module becomes weaker as the reversal lasts longer (*Figure 7D*). Consequently, the membrane potential moves towards the excitation threshold $x_{th}$ to potentiate transition, allowing the analytical expression for $r_2$, *Equation 3*, to become time-dependent. Our hypothesis is consistent with the decay of the glutamate sensor signal on RIB neurites upon AIB stimulation (*Figure 4E*), an observation that may be explained by a depletion of available vesicles for release at the presynaptic site. Note that calcium activity in AIB cell body, like that during a reversal (*Figure 2B*), kept increasing during persistent optogenetic stimulation (*Figure 4—figure supplement 2B*), arguing against the possibility that an opsin-mediated membrane depolarization in the presynaptic neuron undergoes depression upon continuous light activation.

By incorporating the exponential decay of inhibitory synaptic strength (*Equation 1*), we found that the functional form of the transition rate (see Appendix) can be approximated by

$$r_2(t) \approx \frac{k}{\pi} erfi^{-1}\left(\alpha + \beta e^{-t/\tau_g}\right), \tag{4}$$

where $\tau_g = 0.9\,s$ is the decay constant of the glutamate signal (*Figure 4E*). The experimentally measured type-II transition rate is well fit by *Equation 4*.

## Discussion

Complex motor behaviors arise from continual selection and transition among a number of motor primitives. Classic synaptic chain models, in which stereotyped motor sequences arise from feedforward excitation between different groups of neurons, are thought to underlie several motor behaviors such as Zebra Finch singing (*Long et al., 2010*). A feedforward synaptic chain may underlie the replay of spatiotemporal activity patterns in hippocampus during sleep (*Louie and Wilson, 2001*; *Skaggs and McNaughton, 1996*), and generate temporally precise firing patterns that correspond to different actions in the motor cortex of behaving monkeys (*Shmiel et al., 2006*). Alternatively, when several mutually inhibited modules are co-activated by sensory inputs, motor sequences could also emerge by a winner-take-all strategy, a proposed mechanism for the grooming behavior in *Drosophila* (*Seeds et al., 2014*). In mice, mutually inhibitory neurons in the central amygdala have been shown to regulate dimorphic defensive behaviors — flight or freezing — triggered by looming visual stimuli (*Fadok et al., 2017*). Here, we find the two schemes are likely integrated by the *C. elegans* nervous system to generate robust and flexible motor sequences (*Figure 7D*).

In *C. elegans*, feedforward excitation between the backward module and the turning module (*Figure 7A*) can reliably trigger an omega turn followed by forward movement through strong and persistent activation of local interneurons AIB (*Figure 2D* and *Figure 2—figure supplement 1D*). In other words, the action in a motor sequence can be selected through feedforward excitation, triggered by either external sensory stimulus or fluctuations of internal circuit dynamics. The timing of an action can be tuned by augmenting the feedforward excitation with glutamatergic feedforward inhibition between AIB and RIB (*Figure 7A*), and likely by modulating the strength of inhibitory inputs through short term synaptic plasticity. Previously, a tyraminergic feedforward inhibition (*Alkema et al., 2005*; *Pirri et al., 2009*) from the RIM interneurons in the backward module to the SMD motor neurons in the turning module was shown to suppress head movement during reversals. We propose that these functional motifs — feedforward excitation and inhibition — are combined with a nonlinear activation of turning neurons (*Figure 7D*) to produce flexible type-II (RT) transitions.

A simple synaptic chain model predicts that abolishing neural activity in a downstream module would not directly affect upstream neural output. However, when RIV/SAA/SMB in the turning module were ablated or inhibited (*Figure 6B–C*), we observed prolonged reversals during escape

responses. Hence, the turning module may provide feedback inhibition onto the backward module, contributing to the reversal termination during the type-II transition. The cellular and molecular mechanisms for inhibitory feedback remain to be identified. One possible implementation is cholinergic synaptic outputs from SAAD onto RIM and AVA interneurons in the backward module; another possibility is synaptic outputs from RIB onto AVA/AVE (*Figure 7A*). Together, the feedforward coupling between the backward module and the turning module facilitates a defined sequential activity pattern, whereas the winner-take-all operation through mutual inhibition between the two modules avoids an action conflict.

Sensorimotor transformation depends on the initial condition of the network state (*Remington et al., 2018a*; *Remington et al., 2018b*). We show that when the backward motor state is suppressed via the hyperpolarization of interneurons AIB, an identical mechanosensory stimulus is less likely to elicit an escape response (*Figure 2E*). A recent study also demonstrated that mechanosensory stimuli were unlikely to drive other motor programs when *C. elegans* was executing a turn (*Liu et al., 2018*). We propose that the inhibition from the turning module to the backward module (*Figure 7D*) may account for this observation.

We view omega turn, a motor state encoded by transient activity in RIV, as a special manifold connecting two attractors represented by persistent activity in the forward or backward module (*Figure 7D*). Our finding — a combination of feedforward excitation and mutual inhibition between motor states — suggests a new way to control nonlinear dynamics towards a different fixed point (*Morrison et al., 2020*). In our simplified model, neurons within a module were treated as a homogeneous population. Nevertheless, interneurons with heterogeneous functional properties have been found. For example, laser ablation of AIB and RIM in the backward module (*Figure 2A*) differentially affect the probability of spontaneous reversals (*Gray et al., 2005*). While RIM showed increased calcium activity during reversals (*Figure 2—figure supplement 1B*) and promoted reversal upon optogenetic activation (*Figure 2—figure supplement 1E–G*), they were less important in modulating the type-II transition than AIB did (*Figure 2—figure supplement 1D–G*). The impact of functional heterogeneity on the attractor dynamics and motor state transitions remains to be understood.

Our biophysical model suggests that noises in a neural circuit (*Equation 2*) contribute to behavior variability. We speculate that stochasticity in neural dynamics and behaviors may allow animals to efficiently explore the action space (*Dhawale et al., 2017*; *Duffy et al., 2019*; *Tumer and Brainard, 2007*); learning, by which functional connectivity between motor modules is modified through synaptic plasticity, may optimize action selection and timing (*Sutton and Barto, 2017*). We found that in *C. elegans*, the functional connectivity between motor modules consists of feedforward excitation and mutual inhibition. Conserved network motifs may be distributed among mammalian forebrain and midbrain circuits (*Fadok et al., 2017*; *Klaus et al., 2019*). Other animals could use similar algorithms to organize neuronal activities into sequential states to drive motor primitives, by which stereotyped and flexible behaviors emerge.

# Materials and methods

**Key resources table**

| Reagent type (species) or resource | Designation | Source or reference | Identifiers | Additional information |
|---|---|---|---|---|
| Strain, strain background (*C. elegans*) | *C. elegans* strains used and generated in this study | *Caenorhabditis* Genetics Center (CGC) and this paper | | *Supplementary file 1* |
| Recombinant DNA reagent | Plasmids generated in this study | This paper | | *Supplementary file 2* |
| Recombinant DNA reagent | Primers for genes (or promoters) used in this study | This paper | | *Supplementary file 3* |
| Software, algorithm | MATLAB | Mathworks | RRID:SCR_001622 | https://www.mathworks.com |
| Software, algorithm | CoLBeRT system | Samuel Lab | | http://colbert.physics.harvard.edu/ |

*Continued on next page*

*Continued*

| Reagent type (species) or resource | Designation | Source or reference | Identifiers | Additional information |
|---|---|---|---|---|
| Software, algorithm | ImageJ | Media Cybernetics | RRID:SCR_003070 | https://imagej.net/ |
| Software, algorithm | LabVIEW | National Instruments | RRID:SCR_014325 | http://www.ni.com |

### *C. elegans* strains

*C. elegans* strains including wild-type (N2), mutants, and transgenic worms were grown and cultivated according to standard procedures (*Brenner, 1974*). All strains used in this paper can be found in the *Supplementary file 1*. Transgenic worms for optogenetic experiments were cultivated in dark on NGM plates with OP50 bacteria and 0.4 mM all-trans retinal (ATR) for over 5 hr. We used young adult hermaphrodites to perform optogenetic and calcium imaging experiments, and L4 hermaphrodites to obtain expression patterns.

### Molecular biology

Standard molecular biology methods were used. Details of plasmids, promoters and rescue genomic DNA (or cDNA) sequences can be found in *Supplementary file 2–3*.

### Optogenetics

Worms were first washed in M9 buffer (or transferred onto an unseeded NGM plate for 1–3 min), then transferred onto a fresh agar plate [~ 0.8% (w/v) agar in M9 buffer, without food], mounted on a motorized stage. Worms were left to freely explore the new environment for 3–5 min before testing, and were automatically tracked and retained within the field of view of a 10 × objective (Nikon Plan Apo, NA = 0.45) mounted on an inverted microscope (Nikon Ti-U, Japan) via dark field infrared illumination. Worm behaviors were recorded by a CMOS camera (Basler, aca2000-340kmNIR, Germany). MATLAB custom software (MathWorks, Inc Natick, MA, USA) was used for post-processing behavioral data and extracting moving directions and the kinematics of omega turns.

For freely roaming worms without optogenetic stimulation, we only recorded them for 5–8 min.

For worms with optogenetic stimulation, lasers and a digital micromirror device (DLI4130 0.7 XGA, Digital Light Innovations, TX, USA) were used to generate a defined spatiotemporal illumination pattern (*Leifer et al., 2011*) at a specific wavelength (473 nm, 561 nm or 635 nm), and to manipulate the activities of neurons expressing light-activated channels (ChR2, Arch or Chrimson) (*Husson et al., 2012*; *Nagel et al., 2005*). To eliminate the effect of adaptation, single worm was stimulated 5–8 times with at least a 50 second inter-stimulus interval. For example:

1. To trigger escape responses, worms received 1.5 s blue light (short enough and over 80% trials showed responses in pilot experiments) to activate mechanosensory neurons ALM/AVM. All worms in the dataset had at least 20% probability to perform either type-I or type-II transitions. In other cases, 7 s (or 12 s) optogenetic illumination were used to ensure persistent activation/inhibition of local interneurons.
2. In some experiments, worms received sequential optogenetic stimulations with different colors and varying durations controlled by diaphragm shutters (GCI 7102M, Daheng Optics, China). For example, the 1.5 s blue light optogenetic stimulation (to trigger escape response) was followed by green light with a duration of 3–12 s to inhibit other interneurons in the motor control circuit.
3. We also performed selective optogenetic manipulation of interneurons when their cell body positions were sufficiently apart, given that the lateral resolution of our CoLBeRT system is up to ~ 5 µm. For example, RIV and SAAD neurons are separated by at least 20 µm along the dorsal-ventral axis (*Figure 6A*). In order to inhibit SAAD or RIV independently (P*lim-4*::Arch::GFP), we generated a spatial pattern to selectively illuminate the dorsal or ventral side (*Figure 6B* right). At the same time, we monitored GFP emission signals from these neurons excited by the laser (473 nm) to ensure we targeted the correct region (*Figure 6B*).

### Calcium imaging

Calcium imaging was conducted on worms expressing a GCaMP6::wCherry (or mCardinal) fusion protein. Calcium activity was measured as a ratiometric change. For example, neural activity of AIB

was measured as a fluorescence ratio of GFP to RFP ($\Delta R(t)/R_0$) (GCaMP6/wCherry), where $R_0$ is the baseline ratio. In some cases, we define a new normalized ratiometric measure, $\Delta\Re(t) = [\Delta R(t)- \Delta R(t = 0)]/ R_0$.

When cell-specific promoters are available, including AIB and RIB, we performed calcium imaging using wide-field fluorescence microscopy. Unrestrained behaving worms were placed on fresh agarose plates [2% (w/v) agarose in M9 buffer], tracked by a motorized stage using the CoLBeRT system with infrared light illumination (*Leifer et al., 2011*). Blue and green lights were employed to excite GCaMP6 and wCherry (or mCardinal) proteins. Green and red emission signals were captured by a 10 × objective (Nikon Plan Apo, WD = 4 mm; NA = 0.45, Japan) at 50 fps with an exposure time of 20 ms, separated by a dichroic mirror, relayed by an optical splitter (OptoSplit II, Cairn-Research, UK), and projected onto one-half of a sCMOS sensor (Andor Zyla 4.2, UK) simultaneously. Green and red channels were aligned and processed by custom-written MATLAB scripts (*Xu et al., 2018*). Single worm was recorded for 3–10 min.

To image RIV neurons, which lack cell-specific promoters, we picked transgenic worms (P*lim-4*:: GCaMP6::wCherry) with stronger fluorescence expression on RIV, and increased the exposure time (up to 50 ms) to obtain high signal-to-noise ratio images.

## Multi-neuron calcium imaging in a freely behaving worm

To image calcium activity of multiple neurons in a freely behaving worm (e.g., *Figure 2—figure supplement 1B*), we combined a spinning disk confocal inverted microscope (Nikon Ti-U and Yokogawa CSU-W1, Japan) for calcium imaging with a customized upright light path for worm tracking and behavior recording. A worm was placed on an agarose pad [~ 2% (w/v)] mounted on a motorized stage. We used a 40 × air objective (Nikon Plan Apo, WD 0.25–0.17 mm; NA = 0.95, Japan) or a 60 × water immersion objective (Nikon Plan Apo, WD 0.22 mm; NA = 1.20, Japan). The wavelengths of confocal excitation lasers were 488 nm and 561 nm; the emission lights were split (Andor Optosplit II, UK) in front of a sCMOS camera (Andor Zyla 4.2, UK). At the same time, we utilized a customized light path that was aligned to the same $z$ axis to track the worm and to record behavioral data. In the upright path, a low magnification 10 × objective (Nikon Plan Fluor, WD 16 mm; NA = 0.30, Japan) was used to gather fluorescent light excited by confocal lasers, and the fluorescent signal was processed to identify worm positions. A real time feedback signal was sent to a motorized stage to keep the worm head within the center of field of view. Meanwhile, we illuminated the worm by an infrared ring surrounding the high magnification objective to record worm behavior through the upright light path.

## Simultaneous optogenetic manipulation and calcium imaging

To combine calcium imaging and optogenetic manipulation (e.g., *Figure 3B*), a 635 nm laser was added to activate Chrimson. Because blue light can also activate Chrimson, excitation light for GCaMP6 imaging and red light for optogenetic stimulation were synchronized using a TTL signal (LabJack Corp., U3-HV) controlled by LabVIEW (National Instruments Corp., USA). For example, to verify the connectivity between AIB and RIV, we activated AIB using red laser (635 nm) and recorded the calcium signal in RIV using blue LED excitation (M470L3-C1; Thorlabs, USA) for 7 s simultaneously. Both restrained and freely moving worms were tested. Restrained worms were placed on 10% (w/v) agarose plates with coverslips, whereas freely moving ones were placed on 2% (w/v) fresh agarose plates. Single worm was stimulated 5–8 times with at least a 50 s inter-trial interval.

## Thermally-induced escape responses

In addition to optogenetic stimulation of mechanosensory neurons, we also used a thermal stimulus to trigger escape responses. We illuminated the head of a worm with a focused infrared laser light (1480 nm; spot diameter ~120 μm, $\Delta T \sim 2$ °C) for 0.75 s (short enough and over 80% trials showed responses in pilot experiments), and animals responded with reversals or omega turns to avoid the stimulus (*Mohammadi et al., 2013*). The transition rates were qualitatively similar to ALM/AVM-induced escape responses (*Figure 1C–D*). Identical experimental protocols were used when thermally-induced escape responses were followed by optogenetic manipulation or calcium imaging of interneurons.

## Glutamate imaging

Glutamate imaging was conducted on worms expressing iGluSnFR (*Marvin et al., 2013*) on RIB and Chrimson on AIB (*Figure 4E* and *Figure 4—figure supplement 2A*). All worms were restrained on fresh 10% (w/v) agarose plates with coverslips. We used 60 × water immersion objective. Both processes and cell body of RIB were imaged on the same focal plane. Imaging acquisition rate is 150 fps. Like calcium imaging, blue excitation light (weak enough to reduce the bleaching effect; M470L3-C1; Thorlabs) for glutamate imaging and red light (635 nm, 6.11 mW/mm$^2$) for optogenetic stimulation were synchronized with a TTL signal. Glutamate signaling resulted in a change of the green fluorescence signal on the membrane of RIB interneurons. Stimulation and imaging sustained for more than 8 s. Single worm was stimulated 2–3 times with at least a 50 s inter-stimulus interval.

## Optogenetic ablation

We used miniSOG (mini Singlet Oxygen Generator) to ablate specific neurons in *C. elegans*. Upon blue light stimulation, a mitochondria-targeted miniSOG (TOMM20-miniSOG) (*Qi et al., 2012*) or a membrane-targeted miniSOG (PH-miniSOG) (*Xu and Chisholm, 2016*) were employed to induce cell death in cell-autonomous manner. L2/early L3 worms were transferred onto an unseeded NGM plate, restricted by a small ring of filter paper soaked with 100 µM CuCl$_2$. Worms were illuminated for 40–60 min (TOMM20-miniSOG) or 2 min (PH-miniSOG) with blue LED (M470L3-C5; Thorlabs) at the intensity of 0.46 mW/mm$^2$. After illumination, worms were transplanted back to OP50-seeded NGM plates, and were allowed to recover for 1–2 days before behavior testing.

## Behavioral assays and analysis

Recorded movies, in which worm body centerlines were extracted in real time, were further processed semi-automatically in MATLAB to identify locomotor states (forward locomotion, backward locomotion, pause and turn) and other statistical parameters. We also set up a Graphic User Interface (GUI) that allows human interference and proofreading.

Reversal duration was defined as the time from reversal start to reversal end. If one stimulus triggered several reversals, we always scored the first one. Omega turns were identified by either the head touching the body (or tail) or $\theta > 135°$ within a single head swing (*Figure 1—figure supplement 1A*). The end of a turn was identified when a worm opened its coiled posture and began to move forward.

## Transition rate calculation

Transition rates are calculated by the following equations

$$r^i_{forw} = \frac{n^i_{forw}}{\Delta t \cdot (S^i_{forw} + S^i_{turn})}$$

$$r^i_{turn} = \frac{n^i_{turn}}{\Delta t \cdot (S^i_{forw} + S^i_{turn})}$$

We group trials into time bins. For example, here we use time bin $\Delta t = 1s$ for illustration. Let $n^i_{forw}$, $n^i_{turn}$ denote the number of trials that end with type-I or type-II transition in the i-th time bin. $n^1_{forw} = 8$ means there are 8 trials which terminate its reversal with forward movement from 0.0 s to 1.0 s; $n^4_{turn} = 12$ means 12 trials terminate its reversal with a turn from 3.0 s to 4.0 s. Next, we shall use $S^i_{forw}$, $S^i_{turn}$ to represent the number of trials among type-I or type-II transition which survive to the start of the i-th time bin. Naturally, we have $S^i_{forw} = \sum_{j=i}^{\infty} n^j_{forw}$, $S^i_{turn} = \sum_{j=i}^{\infty} n^j_{turn}$. For example, $S^1_{turn} = \sum_{j=1}^{\infty} n^j_{turn}$ is the total number of trials that execute a turn. $S^3_{forw} = \sum_{j=3}^{\infty} n^j_{forw}$ represents the number of all trials among type-I transition that survive to 2.0 s.

## Theoretical account of transition rates and model details

Detailed description can be found in Appendix.

## Quantification and statistical analysis

Quantification and statistical parameters were indicated in the legends of each figure, including the statistical methods, error bars, $n$ numbers (see *Supplementary file 4* for more details), and $p$ values. We applied Mann–Whitney U test, $\chi^2$ test or Fisher's exact test among samples, two-way ANOVA to determine the significance of difference between groups for two factors, and Kolmogorov-Smirnov test to compare probability distributions from two samples. All multiple comparisons were adjusted using Bonferroni correction. We considered $p$ values of $< 0.05$ significant. All analyses were performed using MATLAB.

## Data availability

Raw data of calcium imaging experiments and all code used for modeling or figure generation are available for download from https://github.com/Wenlab/Worm-Motor-Sequence-Generation (*Xin et al., 2020*; copy archived at https://github.com/elifesciences-publications/Worm-Motor-Sequence-Generation). Source data files have been provided for main figures.

## Acknowledgements

We thank Louis Tao and Shangbang Gao for discussions, Pinjie Li for analyzing the reorientation angles during the type-I and type-II transitions. This work was funded by Key Area R&D Program of Guangdong Province with Grant No. 2018B030338001, the Strategic Priority Research Program of the Chinese Academy of Sciences (Pilot study, grant XDPB10, Grant No. XDB39000000), National Science Foundation of China Grants NSFC-31471051 and NSFC-91632102, Key scientific technological innovation research project by Ministry of Education, the Fundamental Research Funds for the Central Universities (QW), and the Canadian Institute of Health Research Foundation Scheme 154274 (MZ). Mei Zhen thanks her lab members Yangning Lu, Maria Lim, and Jyothsna Chitturi for sharing their unpublished optogenetic methods, observations, and reagents used in this study.

## Additional information

### Funding

| Funder | Grant reference number | Author |
| --- | --- | --- |
| Key Area R&D Program of Guangdong Province | 2018B030338001 | Yuan Wang<br>Xiaoqian Zhang<br>Jing Huo<br>Tianqi Xu<br>Quan Wen |
| Chinese Academy of Sciences Strategic Priority Research Program | XDPB10, XDB39000000 | Yuan Wang<br>Xiaoqian Zhang<br>Qi Xin<br>Jing Huo<br>Tianqi Xu<br>Yu Xie<br>Quan Wen |
| National Science Foundation of China | NSFC-31471051 | Yuan Wang<br>Xiaoqian Zhang<br>Qi Xin<br>Jing Huo<br>Tianqi Xu<br>Yu Xie<br>Quan Wen |
| National Science Foundation of China | NSFC-91632102 | Yuan Wang<br>Xiaoqian Zhang<br>Qi Xin<br>Jing Huo<br>Tianqi Xu<br>Yu Xie |
| CIHR | Foundation Scheme 154274 | Wesley Hung<br>Mei Zhen |

| Fundamental Research Funds for the Central Universities | Quan Wen |
|---|---|

The funders had no role in study design, data collection and interpretation, or the decision to submit the work for publication.

## Author contributions

Yuan Wang, Xiaoqian Zhang, Resources, Data curation, Formal analysis, Investigation, Visualization, Writing - original draft, Writing - review and editing; Qi Xin, Data curation, Software, Formal analysis, Visualization, modelling, Writing - original draft; Wesley Hung, Mei Zhen, Resources, Writing - review and editing; Jeremy Florman, Tianqi Xu, Yu Xie, Mark J Alkema, Resources; Jing Huo, Resources, Investigation; Quan Wen, Conceptualization, Supervision, Funding acquisition, Investigation, Writing - original draft, Project administration, Writing - review and editing

## Author ORCIDs

Yuan Wang (iD) https://orcid.org/0000-0002-0471-5754
Xiaoqian Zhang (iD) https://orcid.org/0000-0001-5636-7950
Qi Xin (iD) https://orcid.org/0000-0002-5748-8057
Jeremy Florman (iD) http://orcid.org/0000-0001-7578-3511
Yu Xie (iD) http://orcid.org/0000-0003-3624-0252
Mark J Alkema (iD) http://orcid.org/0000-0002-1311-5179
Mei Zhen (iD) http://orcid.org/0000-0003-0086-9622
Quan Wen (iD) https://orcid.org/0000-0003-0268-8403

## Decision letter and Author response

Decision letter https://doi.org/10.7554/eLife.56942.sa1
Author response https://doi.org/10.7554/eLife.56942.sa2

## Additional files

### Supplementary files

- Supplementary file 1. Strains information.
- Supplementary file 2. Associated plasmids information.
- Supplementary file 3. Associated promoter and gene information.
- Supplementary file 4. Sample size and replicate number for all experiments.
- Transparent reporting form

### Data availability

All data generated or analysed during this study are included in the manuscript.

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

## Appendix 1

This note provides a theoretical account of the time-dependent type-II (RT) transition rate. First, we shall consider how the type-II transition arises through a rapid change of neuronal activity, represented experimentally by inter/motor neurons RIV. Second, we shall argue that the introduction of short-term synaptic depression (STD) in the inhibitory synapses may provide an explanation of the type-II transition rate.

### A biophysical model

Consider a model neuron (e.g. RIV) in the turning module receiving excitatory inputs through gap junction and inhibitory inputs through chemical synapses (*Figure 7A*). The following equation describes the membrane potential $x$ of a neuron determined by the gap junction conductance, inhibitory synaptic conductance and leaky conductance, represented by $G^g$, $G^i$ and $G^c$ respectively (*Varshney et al., 2011*).

$$C\dot{x} = -G^g(x - E^e) - G^i(x - E^i) - G^c(x - E^c) + I^{Noise} \tag{1.1}$$

Since the excitatory gap junction input is from AIB, here we use $E^e$ to represent the voltage of AIB in the first term. $E^i$ and $E^c$ denote the reversal potential of inhibitory synapse and the resting potential. $I^{noise}$ is Gaussian white noise, which could be regarded as the fluctuation of synaptic currents and other sources of noises that are not explicitly considered in the model.

For simplicity, let us first assume that all the inputs, $G^g$, $G^i$ and $G^c$, are constants. This assumption will be relaxed later when we introduce short-term synaptic plasticity in inhibitory synapses, with which $G^i$ decreases while $G^g$ and $G^c$ remain unchanged. $E^e$, the membrane potential of AIB is treated as a constant, provided that the backward module exhibits persistent activity during a reversal. The fixed point $x_0$ is given by

$$x_0 = \frac{E^e G^g + E^i G^i + E^c G^c}{G^g + G^i + G^c} = \left(\sum_m E^m G^m\right) / \sum_m G^m \tag{1.2}$$

where $m$ denotes all kinds of inputs, including gap junctions, chemical synapses and leaky channels. $G^m$ and $E^m$ are the conductance of an input and the corresponding target voltage.

With small perturbation $x_0 \rightarrow x_0 + \Delta x$, we make an approximation

$$C\dot{x} = -\sum_m G^m \cdot \Delta x \tag{1.3}$$

provided that the synaptic inputs from other neurons are insensitive to small fluctuation of the membrane potential of a postsynaptic neuron. In this case, the membrane potential can be viewed as a one-dimensional Brownian particle which obeys the following equation

$$\dot{x} = -k(x - x_0) + \eta, x < x_{th} \tag{1.4}$$

where $k = \sum_m G^m / C$, and $\eta$ is gaussian white noise. We assume that the membrane potential $x$ fluctuates around $x_0$ during a reversal. Once the noise drives the membrane potential across a threshold $x_{th}$, the system transits to a different state: RIV neurons become rapidly depolarized due to a nonlinear rectified activation (*Figure 3E* and *Figure 7B*), starting a turn by activating ventral muscles and terminating the reversal via feedback inhibition (*Figure 7A and D*).

### Calculating the stationary probability density near fixed point with Fokker-Plank equation

Fokker-Plank equation that describes the probability density of the particle position (*Hänggi et al., 1990*) is

$$\frac{\partial P(x,t)}{\partial t} = \frac{\partial}{\partial x}[-k(x-x_0)P(x,t)] + \sigma^2\frac{\partial^2 P(x,t)}{\partial x^2} \tag{2.1}$$

where $\sigma$ determines the amplitude of noise. According to Einstein's theory on Brownian motion, $\sigma$ and the variance of Gaussian white noise $\eta$ has the following relation:

$$\sigma^2 = \frac{\Delta t}{2}\langle\eta^2\rangle \tag{2.2}$$

where $\Delta t$ is the time step.

The stationary solution for Fokker-Planck equation $\frac{\partial P(x,t)}{\partial t}=0$ is given by:

$$P(x) \sim e^{-\frac{k(x-x_0)^2}{2\sigma^2}} \tag{2.3}$$

which is similar to Boltzmann distribution, where $\frac{k(x-x_0)^2}{2}$ can be viewed as the potential energy $U(x)$, and $\sigma^2$ plays the role of temperature $k_B T$.

## Biophysical understanding of the type-II transition

Now consider a terminal threshold $x_{th}$, beyond which the system transits to a different state. Assuming that the barrier is high and the particle is restricted in the energy well, we shall next calculate the small, albeit not negligible probability current $J$ crossing the barrier.

From Fokker-Planck *Equation 2.1* we have:

$$\begin{aligned}\frac{\partial P(x,t)}{\partial t} &= \frac{\partial}{\partial x}[k(x-x_0)P(x,t)] + \sigma^2\frac{\partial^2 P(x,t)}{\partial x^2}\\ &= \frac{\partial}{\partial x}\left[k(x-x_0)P(x,t) + \sigma^2\frac{\partial P(x,t)}{\partial x}\right]\\ &= -\frac{\partial J(x,t)}{\partial x}\end{aligned} \tag{3.1}$$

Current $J$ can be rewritten in another form

$$J(x,t) = -\sigma^2 e^{-\frac{k(x-x_0)^2}{2\sigma^2}}\frac{\partial}{\partial x}\left[e^{\frac{k(x-x_0)^2}{2\sigma^2}}P(x,t)\right] \tag{3.2}$$

Considering a stationary state, we have $\frac{\partial P(x,t)}{\partial t}\approx 0$, $J\approx const$

$$\frac{\partial}{\partial x}\left[e^{\frac{k(x-x_0)^2}{2\sigma^2}}P(x)\right] = -\frac{J}{\sigma^2}e^{\frac{k(x-x_0)^2}{2\sigma^2}} \tag{3.3}$$

Integrate *Equation 3.3* between $x_0$ and $x_{th}$

$$\left[e^{\frac{k(x-x_0)^2}{2\sigma^2}}P(x)\right]\Big|_{x_0}^{x_{th}} = -\frac{J}{\sigma^2}\int_{x_0}^{x_{th}}e^{\frac{k(x-x_0)^2}{2\sigma^2}}dx \tag{3.4}$$

Since the particle is restricted around $x_0$, $P(x_{th})\approx 0$, then we have

$$J = \frac{\sigma^2 P(x_0)}{\int_{x_0}^{x_{th}}e^{\frac{k(x-x_0)^2}{2\sigma^2}}dx} \tag{3.5}$$

To get $P(x_0)$ in *Equation 3.5*, note that the probability of finding the particle around $x_0$ should be 1, and $P(x)=P(x_0)e^{-\frac{k(x-x_0)^2}{2\sigma^2}}$ according to *Equation 2.3*, we have

$$P(x_0)\int_{-\infty}^{+\infty}e^{-\frac{k(x-x_0)^2}{2\sigma^2}}dx = 1 \tag{3.6}$$

Let $A$ denote $\sqrt{\frac{k}{2\sigma^2}}$

$$P(x_0) \int_{x_0-h}^{x_0+h} e^{-\frac{k(x-x_0)^2}{2\sigma^2}} dx = P(x_0) \int_{-h}^{h} e^{-(Ax)^2} dx = P(x_0) \frac{\sqrt{\pi}}{A} erf(Ah) = 1 \tag{3.7}$$

where $erf(x) = \frac{2}{\sqrt{\pi}} \int_0^x e^{-z^2} dz$ is the error function.

By taking the limit $h \to +\infty$, $erf(Ah) \to 1$, we find

$$P(x_0) = \frac{A}{\sqrt{\pi}} \tag{3.8}$$

The denominator $\int_{x_0}^{x_{th}} e^{\frac{k(x-x_0)^2}{2\sigma^2}} dx$ in **Equation 3.5** can be calculated directly

$$\int_{x_0}^{x_{th}} e^{\frac{k(x-x_0)^2}{2\sigma^2}} dx = \int_0^{x_{th}-x_0} e^{(Ax)^2} dx = \frac{\sqrt{\pi}}{2A} erfi(A(x_{th}-x_0)) \tag{3.9}$$

where $erfi(x) = \frac{2}{\sqrt{\pi}} \int_0^x e^{z^2} dz$ is imaginary error function

Substituting **Equations 3.8– 3.9** into **Equation 3.5**, we finally obtain the transition rate $r$

$$r = J = \frac{k}{\pi} \cdot erfi^{-1}\left(\sqrt{\frac{k}{2\sigma^2}}(x_{th}-x_0)\right) \tag{3.10}$$

## Transition rate with short-term synaptic depression

Since the membrane time constant of a neuron (~10 ms) is much smaller than the behavioral timescale (~s), the probability density approaches the stationary state rapidly and **Equation 3.10** is a constant rather than a function of time, resulting in an exponential distribution of the survival function. To explain the time-dependent type-II transition rate (**Figure 1C–D**), we introduce plasticity in the inhibitory synapses, analogous to short-term synaptic depression (STD).

As the inhibition onto the model neuron becomes weaker, fixed point $x_0$ moves towards threshold $x_{th}$; $k$, which characterizes the steepness of the potential energy, becomes shallower. Both analytical calculation and simulation suggest that the change of $x_0$ has a much larger effect on the transition rate in the region $x_{th} - x_0 \ll E^e - E^i$.

In this regime, consider a simple case when $x_0$ varies with time for a fixed $k$. Assuming the strength of synaptic inhibition decreases exponentially, we have

$$
\begin{aligned}
r_2(t) &= \frac{k}{\pi} \cdot erfi^{-1}\left(\sqrt{\frac{k}{2\sigma^2}}(x_{th}-x_0(t))\right) \\
&= \frac{k}{\pi} \cdot erfi^{-1}\left(\sqrt{\frac{k}{2\sigma^2}}\left(x_{th}-x_{fin}+(x_{fin}-x_{ini}) \cdot e^{-t/\tau}\right)\right) \\
&= \frac{k}{\pi} \cdot erfi^{-1}\left(\alpha + \beta \cdot e^{-t/\tau}\right)
\end{aligned}
\tag{4.1}
$$

**Equation 4.1** was used to fit the type-II transition rate.

