## [Decision Letter]

**Acceptance summary:**

To execute competent behaviours, animals have to select, time and orchestrate their individual actions into orderly action sequences. By studying the escape response of *C. elegans*, Wang and colleagues provide intriguing insights into the underlying neuronal circuit mechanisms. While excitatory electrical signalling between an interneuron and a motoneuron ensures the sequential execution of actions, mutual inhibition via chemical synapses affects variable choices and timing. Such a combined excitatory feed-forward and winner-takes-all mutual-inhibition mechanism might be a general principle by which larger organism also organise more complex action sequences.

**Decision letter after peer review:**

Thank you for submitting your article "Flexible Motor Sequence Generation during Stereotyped Escape Responses" for consideration by *eLife*. Your article has been reviewed by three peer reviewers, one of whom is a member of our Board of Reviewing Editors, and the evaluation has been overseen by Piali Sengupta as the Senior Editor. The reviewers have opted to remain anonymous.

The reviewers were very enthusiastic about your manuscript and this decision has been arrived at following extensive consultation among them. The Reviewing Editor has drafted this decision to help you prepare a revised submission based on the detailed comments below.

As the editors have judged that your manuscript is of interest, but as described below that additional experiments and/or analyses are required before it is published, we would like to draw your attention to changes in our revision policy that we have made in response to COVID-19 (https://elifesciences.org/articles/57162). First, because many researchers have temporarily lost access to the labs, we will give authors as much time as they need to submit revised manuscripts. We are also offering, if you choose, to post the manuscript to bioRxiv (if it is not already there) along with this decision letter and a formal designation that the manuscript is "in revision at *eLife*". Please let us know if you would like to pursue this option.

Summary:

In "Flexible Motor Sequence Generation during Stereotyped Escape Responses", Wang et al. use behavioral analysis, calcium and glutamate imaging, optogenetics and a biophysical model to investigate the circuit mechanisms that generate an action sequence, the escape response of *C. elegans*. First, the authors characterize the motor sequences triggered either by optogenetic activation of mechanosensory neurons ALM/AVM or via a heat stimulus. Using a biophysical model, they classify the escape response into distinct action patterns (forward-reverse-omega turn vs forward-reverse-forward). Then, the authors investigate the role of the AIB interneurons in the transition from reversal to omega turns and find evidence that gap junctions between AIB and inter/motor neuron RIV mediate this transition. The authors then investigate the role of inhibitory glutamate signaling from AIB to RIB, an interneuron upstream of RIV, in shaping the dynamics of the reverse/turn transition. Lastly, they look into the role of turning module neurons (RIV, SAA and SMB) and find some initial evidence supporting that these neurons promote reversal termination. These results strongly support a model for the generation of behavioral sequences: as opposed to a feed-forward excitatory synaptic chain mechanism, the authors propose a mechanism involving concomitant feed-forward excitation and inhibition together with feedback inhibition that enables a more flexible control of the action sequence. This model is very interesting and potentially relevant for our understanding of motor control in general.

Essential revisions:

1) Flexibility of the escape response is observed in two ways: variability in the timing of behavioral motifs, as well as the choice between transitions type I and II; these outcomes seem to be interrelated as was observed in previous work. This concept should be better introduced and made clear throughout the text.

2) The authors describe the escape response with an alternative choice between two discrete and distinct transitions (type I vs type II). Here, they rely on a cutoff at 135 degrees reorientation angle; although this cutoff was used a few times in the past literature, to date it seems largely arbitrary. Since this is a key step in their analyses convincing evidence should be provided that justifies this distinction. Is there a bimodal distributions of reorientation angles, supporting this cutoff, under the present experimental conditions?

Note, that Kaplan et al., 2020 (see Figure S5I) reported such a bimodal distribution but here the cutoff should be around 100 degrees; based on Pierce-Shimomura et al., 1999 (Figure 9) bimodality is less obvious and if at all supporting a cutoff of around 45 degrees. Moreover, Szigeti et al., 2016 makes a strong argument of a continuum in omega turns, hence no subdivision should be made at all. On the other extreme, Broekmans et al., 2016 suggest a third transition (δ turn). In the light of these different findings, the authors need to show whether under their conditions the distribution of post-reversal reorientation angles is indeed bimodal and perhaps determine a more objective cutoff to support such behavioral classification into transitions I (no turn) and II (omega turn).

3) Model: reviewer #1: The authors claim that "the statistics were better described by introducing two types of transitions and the corresponding transition rates *r(t)*." However, no evidence is provided that this model indeed outperforms alternative models, e.g. a single transition rate with a continuum of post reversal reorientation angles. Or a model without plasticity in inhibitory synapses. Please provide additional sophisticated analysis scrutinizing the favoured model against alternative hypotheses.

The biophysical model is largely presented as a black box throughout the main text and little guidance to the general reader is provided, so that one has to work through the supplemental note, which is very difficult for the non-modeling-expert to follow. Which components of the model were crucial to obtain good fits and does the final model really outperforms simpler/alternative models?

Reviewer #2: The model isn't well integrated into the main text. The model is referred to in the Discussion and in relation to Figure 7 but appears to have a tenuous connection at this point. Ultimately, it only provides a fit function for the time-dependent transition rate. However, the Discussion suggests that the authors want to give more weight to the model as a basis for understanding the underlying neural circuit.

There are a few issues with the model as presented in the supplementary note:

- Assumptions are stated but are not justified or tested for their impact on the conclusion. For example, the assumption of the white-noise inputs from other neurons and the ad hoc assumption that the synaptic inhibition decreases exponentially.

- In the end the function has 3 (4?) free parameters to fit a curve. We are not surprised that the fit quality is good, but from the sparse description of the model it is unclear if it adds anything that goes substantially beyond just fitting an error function to the data.

4) At present, this study provides no evidence of neuronal correlates distinguishing transitions type I vs II. Simply triggering activity to reversal starts, reversal ends, and turns has been done in previous literature. The difference in mean amplitudes of AIB activity (type I vs II) could support the continuous model equally and/or be simply a result of different reversal durations. Moreover, we are surprised that it is not shown how the activity of RIV during type I vs. II differ? How is AIB and RIV activity distributed and can one predict type I vs type II just from certain features in AIB/RIV activity?

5) The *inx-1, unc-7, unc-9* triple mutant data are hard to interpret because UNC-7 and UNC-9 are broadly expressed and previous literature shows that these mutants have substantial locomotion defects, thus there could be pleiotropic and additive effects from mutating several innexins. This should also be made clear in the main text, since at present, the authors mention that "several innexin proteins including INX-1, UNC-7 and UNC-9, are reported to express in AIB and RIV inter/motor neurons" but do not mention that especially UNC-7 and UNC-9 are expressed broadly in most other neurons.

Similarly, paragraph two of subsection “Feedforward coupling between the backward and turning modules drives omega turns” read that the *inx-1* mutants still execute omega turns, leading the authors to conclude that multiple innexins are at play. However, it could also be that multiple neurons are at play, this possibility cannot be ruled out at this stage. The reasoning for performing the calcium imaging of RIV during AIB activation in the triple mutants is not clear. Was this experiment performed in the *inx-1* single mutants, which is more specifically expressed in AIB? If so, these data should be shown.

6) There is no consensus on what marks the end of an omega turn. Please provide the definition and justification for this study. Otherwise, Figure 3A and Figure 2—figure supplement 1B are difficult to interpret.

7) One thing that might be nice would be if they could address the issue of stochasticity a bit more in the Discussion. Do the authors have any speculation why the behaviour is not more deterministic? And does their model give any insight into turn and reversal coupling in unstimulated animals undergoing spontaneous reversals?

8) Language:

The type-I /type-II language makes the text hard to read. It would be much easier for readability if the two cases were using an abbreviation that connects to the behaviors, for example type-I could be called RF and type-II – RT. This way the reader doesn't have to remember a somewhat arbitrary assignment of I/II.

9) Introduction: “(…) but a deep connection between theories and experiments remains yet to be established”. What constitutes a “deep connection”? Text suggests the authors are claiming they are providing one, which we don't see based on how little the model is integrated in the main text.

10) Discussion: " Several mechanisms may explain the decay of the iGluSnFR sensor signal, one being a depletion of available vesicles for release at the presynaptic site, analogous to short-term synaptic depression". It would be relevant to name alternative mechanisms here.

11) Discussion: "(…) are thought to underlie several motor behaviors such as Zebra Finch singing" would benefit from more examples and citations.

12) Winner-takes all strategy:

While this aspect can be understood from the data, the winner-takes all strategy should be more explicitly connected to the data in the Results section and explained in the Discussion.

13) Materials and methods:

The extensive documentation of strains, primers, promoters is great, yet it is often unclear which method was used for each figure panel. The Materials and methods section could be improved by a finer substructure with more subsections within similar groupings of experiments, that were performed using different instruments (e.g. optogenetics).

14) For multi-color imaging the Materials and methods section should explain how camera alignment was achieved.

15) Data availability:

We found the code accessible online. The link is included in the transparent reporting form but should also be repeated in the Materials and methods. A statement that custom MATLAB scripts were used is insufficient.

16) The order of neuronal activations involved in the action sequence (forward – reversal – post-reversal turn – forward) showing concomitant activity of AIB with reversal neurons followed by activation of RIV with turning neurons SMD and then RIB was shown already for immobilized worms in Kato et al., 2015. Moreover, ramping AIB activity during reversal has been shown before in Luo et al., 2014; Kato et al., 2015; Laurent et al., 2015, all for freely moving worms. The relationships of interneuron activities with reversal starts in freely moving worms (Figure 2—figure supplement 1B) were all shown already in Kato, 2015 and multiple other studies. It would be fair to credit this work and thereby highlighting better what is really new to this present study.

---

## [Author Response]

Essential revisions:1) Flexibility of the escape response is observed in two ways: variability in the timing of behavioral motifs, as well as the choice between transitions type I and II; these outcomes seem to be interrelated as was observed in previous work. This concept should be better introduced and made clear throughout the text.

Thanks for the suggestion. We have now amended the following sentences in the Introduction:

“Notably, which action to select and when to execute exhibit trial to trial variability, and they can be coupled. For example, a previous study (Gray et al., 2005) has shown that a longer reversal is likely to be followed by an omega turn.”

In the Results section:

“We first examined neuronal correlate of behavioral flexibility in action selection. We compared the AIB ramping activity (P*inx-1*::GCaMP6; P*inx-1*::wCherry) in different action sequences during either spontaneous or thermal-stimulus-triggered behaviors (Figure 2B-C and Figure 2—figure supplement 1B-C).” and “We next investigated behavioral flexibility in the timing of an action.”

2) The authors describe the escape response with an alternative choice between two discrete and distinct transitions (type I vs type II). Here, they rely on a cutoff at 135 degrees reorientation angle; although this cutoff was used a few times in the past literature, to date it seems largely arbitrary. Since this is a key step in their analyses convincing evidence should be provided that justifies this distinction. Is there a bimodal distributions of reorientation angles, supporting this cutoff, under the present experimental conditions?

Thanks for bringing up this important point. We reanalyzed our dataset by computing post-reversal orientation angle between the head and the tail (Figure 1—figure supplement 1A). Intuitively, the head and the tail were diametrically opposed to each other in the type-II transition; whereas they were likely aligned to each other in the type-I transition. Indeed, this was what we found from our dataset and the distribution is apparently bimodal.

Note, that Kaplan et al., 2020 (see Figure S5I) reported such a bimodal distribution but here the cutoff should be around 100 degrees; based on Pierce-Shimomura et al., 1999 (Figure 9) bimodality is less obvious and if at all supporting a cutoff of around 45 degrees. Moreover, Szigeti et al., 2016 makes a strong argument of a continuum in omega turns, hence no subdivision should be made at all. On the other extreme, Broekmans et al., 2016 suggest a third transition (δ turn). In the light of these different findings, the authors need to show whether under their conditions the distribution of post-reversal reorientation angles is indeed bimodal and perhaps determine a more objective cutoff to support such behavioral classification into transitions I (no turn) and II (omega turn).

Our analysis in Figure 1—figure supplement 1A supports the classification. Note that during the escape responses, most turns during the type-II transition were ended with head touching the body or tail (283/322, Figure 1—figure supplement 1A). Our experimental conditions are different from those in Szigeti et al., 2016, which argues for a continuum in omega turns.

3) Model: reviewer #1: The authors claim that "the statistics were better described by introducing two types of transitions and the corresponding transition rates r(t)." However, no evidence is provided that this model indeed outperforms alternative models, e.g. a single transition rate with a continuum of post reversal reorientation angles. Or a model without plasticity in inhibitory synapses. Please provide additional sophisticated analysis scrutinizing the favoured model against alternative hypotheses.

We apologize that the original sentences in the main text may have caused some confusion. There are three reasons that motivate us to introduce two types of transitions. First, the distribution of post-reversal orientation angles is bimodal (Figure 1—figure supplement 1A), as addressed above. Second, the distribution of reversal length is statistically better described by a bimodal distribution (Figure 1—figure supplement 1B, tested using Gaussian mixture model). Third and most importantly, our cell ablation study (Figure 6C) could abolish the type-II transition without affecting the type-I transition. The ablation led to an exponent distribution of the survival function (Figure 6C insets), which was not observed in the statistics of wild type animals. These experimental observations and analysis argue strongly for considering two types of transitions.

We thank the reviewer for asking an alternative model without plasticity in inhibitory synapses. In new Figure 7C, we showed by simulation that the type-II transition rate would be time-independent in the absence of synaptic plasticity. We also explained in the main text why this would be the case.

The biophysical model is largely presented as a black box throughout the main text and little guidance to the general reader is provided, so that one has to work through the supplemental note, which is very difficult for the non-modeling-expert to follow. Which components of the model were crucial to obtain good fits and does the final model really outperforms simpler/alternative models?

We agree with the reviewer that the model should be better integrated with the text. We have outlined the essential ingredients of the model in the Results section.

1) We explain why short-term synaptic plasticity could account for the time-dependent type-II transition rate. We show with both simulation and analytical arguments that an alternative model without introducing synaptic plasticity would lead to a constant transition rate (Figure 7C).

2) We fit the experimentally measured type-II transition rate with r2(t)=k1erfi(k2+k3e−t/τg) , a formula derived from our model. One of the parameters – the decay time constant of synaptic strength τg – was directly obtained from the decay of glutamate sensor signal on RIB upon AIB optogenetic activation.

3) We show that our formula provides a better fit than the standard sigmoid function 1k2+k3e−t/τ with the same *number* of free parameters k2,k3,τ. We think that the introduction of imaginary error function contributes to a better fit of the transition rate (*Figure 7E*).

Reviewer #2: The model isn't well integrated into the main text. The model is referred to in the Discussion and in relation to Figure 7 but appears to have a tenuous connection at this point. Ultimately, it only provides a fit function for the time-dependent transition rate. However, the Discussion suggests that the authors want to give more weight to the model as a basis for understanding the underlying neural circuit.

These questions are similar to those raised by reviewer #1. See above for our responses.

There are a few issues with the model as presented in the supplementary note:- Assumptions are stated but are not justified or tested for their impact on the conclusion. For example, the assumption of the white-noise inputs from other neurons and the ad hoc assumption that the synaptic inhibition decreases exponentially.

We apologize for being not very clear in explaining this point. η could be regarded as fluctuations in synaptic currents (Lindsay, Thiele and Lockery, 2011; Narayan, Laurent and Sternberg, 2011) and other sources of noises that are not explicitly considered in the model. Because we are agnostic to the temporal structure of fluctuations, by the principle of Occam’s Razor, we treated η as Gaussian white noise.

We made a direct connection between the decay of inhibitory synaptic strength and the glutamate fluorescent signal. Glutamate signal can be well fit by an exponential function at long timescale (Figure 4E) and we used the time constant τg as a constraint in our model. In the main text we now stated “Our hypothesis is consistent with the decay of the glutamate sensor signal on RIB neurites upon AIB stimulation (Figure 4E), an observation that may be explained by a depletion of available vesicles for release at the presynaptic site. … ”

- In the end the function has 3 (4?) free parameters to fit a curve. We are not surprised that the fit quality is good, but from the sparse description of the model it is unclear if it adds anything that goes substantially beyond just fitting an error function to the data.

We addressed these questions above.

4) At present, this study provides no evidence of neuronal correlates distinguishing transitions type I vs II. Simply triggering activity to reversal starts, reversal ends, and turns has been done in previous literature. The difference in mean amplitudes of AIB activity (type I vs II) could support the continuous model equally and/or be simply a result of different reversal durations. Moreover, we are surprised that it is not shown how the activity of RIV during type I vs. II differ? How is AIB and RIV activity distributed and can one predict type I vs type II just from certain features in AIB/RIV activity?

Thanks for the questions. Our analysis suggests that the mean amplitude of AIB calcium signal is not a good predictor of whether the animal would execute type-I or type-II transitions. A better predictor of action selection is the ramping rate of calcium fluorescent signal, that is ΔFΔt. The higher the ramping rate of AIB, the more likely the animal will end its reversal with a turn. We have made the text more clearly, see subsection “Local interneurons in the backward module modulate motor state transitions”.

We have also added new calcium imaging experiments of RIV (Figure 3A). RIV calcium activity remained quiescent during the type-I transition and rapidly rose immediately before the type-II transition. These results have also been added to the main text, see subsection “Feedforward coupling between the backward module and the turning module drives the omega turn”.

5) The inx-1, unc-7, unc-9 triple mutant data are hard to interpret because UNC-7 and UNC-9 are broadly expressed and previous literature shows that these mutants have substantial locomotion defects, thus there could be pleiotropic and additive effects from mutating several innexins. This should also be made clear in the main text, since at present, the authors mention that "several innexin proteins including INX-1, UNC-7 and UNC-9, are reported to express in AIB and RIV inter/motor neurons" but do not mention that especially UNC-7 and UNC-9 are expressed broadly in most other neurons.

In order to better explain the rationale of our experimental design, we have rearranged the content of this section. Please see subsection “Feedforward coupling between the backward module and the turning module drives the omega turn”. Briefly, we clarify a few points.

1) In Figure 3B-C, we tested the consequence of connectivity between AIB and RIV. We probe their functional connectivity by activating AIB using Chrimson and observing calcium changes in RIV. The prominent connections between AIB and RIV is gap junction (Figure 2A), and UNC-7, UNC-9 and INX-1 are all reported to be expressed in AIB and RIV. By using the triple mutant, instead of single mutant, the gap junction between these two neurons should be more severely impaired if not completely removed.

Although these innexins are expressed widely in the nervous system, the effect of the loss of gap junctions in other circuits is irrelevant to our analysis: we are testing the effect of direct activation of AIB on RIV activity. The effect of the loss of these innexins severely impairs the increase in RIV activity in response to AIB activation, suggesting that AIB connect to RIV through gap junctions.

We now added: “Several innexin, including INX-1, UNC-7 and UNC-9, have been reported to be expressed in AIB and RIV (Altun et al., 2009; Bhattacharya et al., 2019). […] RIV remained quiescent upon AIB stimulation (Figure 3C red and Figure 3—figure supplement 1D), indicating that gap junction coupling underlies AIB stimulation-mediated RIV calcium activity.”

We also apologize for not clarifying that these innexins are widely expressed. We have amended the text to reflect this notion.

Similarly, paragraph two of subsection “Feedforward coupling between the backward and turning modules drives omega turns” read that the inx-1 mutants still execute omega turns, leading the authors to conclude that multiple innexins are at play. However, it could also be that multiple neurons are at play, this possibility cannot be ruled out at this stage. The reasoning for performing the calcium imaging of RIV during AIB activation in the triple mutants is not clear. Was this experiment performed in the inx-1 single mutants, which is more specifically expressed in AIB? If so, these data should be shown.

2) Our logic for performing calcium imaging experiment in triple innexin mutants, instead of single mutant like the loss of INX-1 alone, is due to high possibility that multiple innexins function redundantly at the gap junction between AIB and RIV. In this case, *inx-1* single mutant would only have a partial behavioral defect in turning variability and restoring INX-1 alone should restore the turning probability (Figure 3D).

However, we agree with the reviewer that these experiments did not exclude the possibility that multiple circuit pathways can trigger turning behaviors. We therefore have amended the text to reflect this notion.

“UNC-7 and UNC*-*9 are broadly expressed in the motor circuit, and *unc-7* or *unc-9* mutants exhibit uncoordinated movements that prohibit them from completing a motor sequence (Barnes and Hekimi, 1997; Brenner, 1974; Kawano et al., 2011; Starich et al., 1993; Xu et al., 2018. […] Because *inx-1* mutants were still capable of generating omega turns, we propose that either multiple innexins between AIB and RIV, or parallel circuit pathways are at play.”

6) There is no consensus on what marks the end of an omega turn. Please provide the definition and justification for this study. Otherwise, Figure 3A and Figure 2—figure supplement 1B are difficult to interpret.

We have amended the definition of turn ends in Behavioral assays and analysis section in Materials and methods:

“The end of a turn was identified when a worm opened its coiled posture and began to move forward.”

7) One thing that might be nice would be if they could address the issue of stochasticity a bit more in the Discussion. Do the authors have any speculation why the behaviour is not more deterministic? And does their model give any insight into turn and reversal coupling in unstimulated animals undergoing spontaneous reversals?

Thank you for the suggestion. We think the internal noise, such as the fluctuation of synaptic currents, may make an important contribution to behavioral variability. We have added:

“Our biophysical model suggests that noises in a neural circuit (Equation 2) contributes to behavior variability. We speculate that stochasticity in neural dynamics and behaviors may allow animals to efficiently explore the action space (Dhawale et al., 2017; Duffy et al., 2019; Tumer and Brainard, 2007); learning, by which functional connectivity between motor modules is modified through synaptic plasticity, may optimize action selection and timing (Sutton and Barto, 2017).”

Regarding turn and reversal coupling in spontaneous movements, we speculate that fluctuation of internal brain dynamics could trigger feedforward excitation between motor modules. We have added:

“In *C. elegans*, feedforward excitation between the backward module and the turning module (Figure 7A) can reliably trigger an omega turn followed by forward movement through strong and persistent activation of local interneurons AIB (Figure 2D and Figure 2—figure supplement 1D). In other words, the action in a motor sequence can be selected through feedforward excitation, triggered by either external sensory stimulus or fluctuations of internal circuit dynamics.”

8) Language:The type-I /type-II language makes the text hard to read. It would be much easier for readability if the two cases were using an abbreviation that connects to the behaviors, for example type-I could be called RF and type-II – RT. This way the reader doesn't have to remember a somewhat arbitrary assignment of I/II.

Thanks for the suggestion. In the main text, we have changed type-I transition to type-I (RF) transition, and type-II transition to type-II (RT) transition.

9) Introduction: “(…) but a deep connection between theories and experiments remains yet to be established”. What constitutes a “deep connection”? Text suggests the authors are claiming they are providing one, which we don't see based on how little the model is integrated in the main text.

We have integrated the model to the Results section.

10) Discussion: " Several mechanisms may explain the decay of the iGluSnFR sensor signal, one being a depletion of available vesicles for release at the presynaptic site, analogous to short-term synaptic depression". It would be relevant to name alternative mechanisms here.

We have revised this sentence.

11) Discussion: "(…) are thought to underlie several motor behaviors such as Zebra Finch singing" would benefit from more examples and citations.

We now added:

“Classic synaptic chain models, in which stereotyped motor sequences arise from feedforward excitation between different groups of neurons, are thought to underlie several motor behaviors such as Zebra Finch singing (Long et al., 2010). A feedforward synaptic chain may underlie the replay of spatiotemporal activity patterns in hippocampus during sleep (Louie and Wilson, 2001; Skaggs and McNaughton, 1996), and generate temporally precise firing patterns that correspond to different actions in the motor cortex of behaving monkeys (Shmiel et al., 2006). ”

12) Winner-takes all strategy:While this aspect can be understood from the data, the winner-takes all strategy should be more explicitly connected to the data in the Results section and explained in the Discussion.

Thanks for the suggestion. In the Results section, we now added:

“Together, our data suggest that the feedforward inhibition (Figure 4) and feedback inhibition (Figure 6) between the backward module and the turning module implement a winner-take-all computation for action selection. The motor module with the highest level of activity stays active by suppressing the activities of other modules.”

In the Discussion section, we now added:

“Together, the feedforward coupling between the backward module and the turning module facilitates a defined sequential activity pattern, whereas the winner-take-all operation through mutual inhibition between the two modules avoids an action conflict.”

13) Materials and methods:The extensive documentation of strains, primers, promoters is great, yet it is often unclear which method was used for each figure panel. The Materials and methods section could be improved by a finer substructure with more subsections within similar groupings of experiments, that were performed using different instruments (e.g. optogenetics).

Thanks. We have revised the Materials and methods section accordingly.

14) For multi-color imaging the Materials and methods section should explain how camera alignment was achieved.

Thanks. We have added details to the calcium imaging section:

“Green and red emission signals were captured by a 10 × objective (Nikon Plan Apo, WD = 4 mm; NA = 0.45, Japan) at 50 fps with an exposure time of 20 ms, separated by a dichroic mirror, relayed by an optical splitter (OptoSplit II, Cairn-Research, UK), and projected onto one-half of a sCMOS sensor (Andor Zyla 4.2, UK) simultaneously. Green and red channels were aligned and processed by custom-written MATLAB scripts (Xu T et al., 2018).”

15) Data availability:We found the code accessible online. The link is included in the transparent reporting form but should also be repeated in the Materials and methods. A statement that custom MATLAB scripts were used is insufficient.

We have added the Data availability to the end of the Materials and methods section.

16) The order of neuronal activations involved in the action sequence (forward – reversal – post-reversal turn – forward) showing concomitant activity of AIB with reversal neurons followed by activation of RIV with turning neurons SMD and then RIB was shown already for immobilized worms in Kato et al., 2015. Moreover, ramping AIB activity during reversal has been shown before in Luo et al., 2014; Kato et al., 2015; Laurent et al., 2015, all for freely moving worms. The relationships of interneuron activities with reversal starts in freely moving worms (Figure 2—figure supplement 1B) were all shown already in Kato, 2015 and multiple other studies. It would be fair to credit this work and thereby highlighting better what is really new to this present study.

Thanks for pointing this out. We have revised and presented a more comprehensive review of earlier works at the beginning of the Results section. We added:

“Structural and functional studies of AIB (Gray et al., 2005; White, Southgate, Thomson and Brenner, 1986) indicate that they may play important roles in motor state transitions (Figure 2A). […] Third, AIB exhibite ramping calcium activity during reversals (Kato et al., 2015; Laurent et al., 2015; Luo et al., 2014), and finally, laser ablation of AIB significantly reduces the frequency of reversals during food search behavior (Gray et al., 2005).”

and

“How do AIB drive turning behaviors? Whole brain imaging in immobilized animals implicated that AIB and their electrically-coupled partners RIV (Figure 2A and Figure 3—figure supplement 2) exhibited sequentially activated patterns (Kato et al., 2015). We compared RIV activity patterns (P*lim-4*::GCaMP6) underlying different motor sequences during spontaneous behaviors.”